# Natural α-Glucosidase and Protein Tyrosine Phosphatase 1B Inhibitors: A Source of Scaffold Molecules for Synthesis of New Multitarget Antidiabetic Drugs

**DOI:** 10.3390/molecules26164818

**Published:** 2021-08-09

**Authors:** Massimo Genovese, Ilaria Nesi, Anna Caselli, Paolo Paoli

**Affiliations:** Department of Experimental and Clinical Biomedical Sciences “Mario Serio”, University of Florence, 50139 Florence, Italy; massimo.genovese@student.unisi.it (M.G.); i.nesi2@student.unisi.it (I.N.); anna.caselli@unifi.it (A.C.)

**Keywords:** PTP1B, α-glucosidase, insulin signaling, drug discovery, type 2 diabetes

## Abstract

Diabetes mellitus (DM) represents a group of metabolic disorders that leads to acute and long-term serious complications and is considered a worldwide sanitary emergence. Type 2 diabetes (T2D) represents about 90% of all cases of diabetes, and even if several drugs are actually available for its treatment, in the long term, they show limited effectiveness. Most traditional drugs are designed to act on a specific biological target, but the complexity of the current pathologies has demonstrated that molecules hitting more than one target may be safer and more effective. The purpose of this review is to shed light on the natural compounds known as α-glucosidase and Protein Tyrosine Phosphatase 1B (PTP1B) dual-inhibitors that could be used as lead compounds to generate new multitarget antidiabetic drugs for treatment of T2D.

## 1. Introduction

Type 2 diabetes is a complex pathology characterized by hyperglycemia and metabolic abnormalities affecting different organs and tissues, such as liver, muscle, adipose tissue, and pancreas. To date, subjects affected by T2D can rely on several oral antihyperglycemic drugs showing different mechanisms of action to keep glycaemia under control. These drugs include: inhibitors of intestinal α-glucosidases, which delay intestinal absorption of glucose; metformin, which blocks hepatic gluconeogenesis; different types of secretagogues that stimulate the release of insulin from pancreatic β-cells; thiazolidinediones, which stimulate the storage of circulating fatty acids into adipocytes, thereby improving insulin sensitivity in several peripheral tissues; and the sodium/glucose cotransporter 2 (SGLT-2) inhibitors, which impair the re-uptake of glucose in the renal tubules [1]. The choice of the most appropriate hypoglycemic drug for a patient depends on several factors, such as the patient’s general condition, the presence of comorbidities, tolerance, and the patient’s response to the drug. Generally, most diabetic patients showing hyperglycemia without further pathological complications respond positively to single-drug based therapy (Figure 1), experiencing a decrease of blood sugar levels and an improvement of their general conditions.

However, many clinical studies showed that benefits obtained with this approach are transient, and in the medium to long term, patients experience a gradual rise in blood sugar and a worsening of general health conditions. In some cases, the up-scaling of drug dosage can allow regaining of the glycemic target, with the hope that, at the same time, no adverse effects related to high doses of the drug occur [2]. The failure of mono-drug therapy is mainly due to the inability of such drugs to replace physiological functions of insulin. Indeed, even if these drugs are able to compensate a specific metabolic defect, they unexpectedly induce severe unbalance in other metabolic pathways.

### 1.1. Combination versus Mono-Drug Therapy for Treatment of T2D

The combination of two or more anti-hyperglycemic drugs acting on different biological targets is a therapeutic option often used for treatment of diabetic patients who do not adequately respond to mono-drug therapy [3]. From a theoretical point of view, the purpose of this strategy is to generate a synergistic effect by acting on different targets involved directly or indirectly in the control of glycemia (Figure 2).

This approach aims to generate a synergistic effect, improving glycemic control using a lower dosage of each drug than the doses provided by mono-drug therapy. However, despite the undoubted advantages of such a pharmacological approach, many clinical trials revealed that multi-drug therapies are difficult to manage for the majority of patients for different reasons. For example, often a fine adjustment of dosages is required, or patients are asked to take medications at different times of the day because of their different pharmacokinetics. These factors make it very difficult for patients to comply with the therapeutic protocol, thus increasing the risk of not reaching the glycemic target. The consequent uncontrolled fluctuations in blood sugar can be deleterious, forcing patients to change treatment to avoid further complications. A practical solution to this problem may arise from taking combinations of oral hypoglycemic drugs in fixed and pre-established doses depending on the desired effects. This strategy reduces the complexity of therapeutic regimen and improves patient adherence to treatment [4]. Overall, the clinical studies carried out so far confirmed that combinatorial therapies imply, in the short to medium term, significant benefits compared to mono-therapy. However, in the long term, the efficacy of this therapeutic approach remains to be confirmed [5]. In conclusion, all evidence suggests that multitargets therapies seem to guarantee a better quality of life for people affected by T2D.

### 1.2. Toward the Multiple Designed Ligands (MDLs) for Treatment of T2D

Studies performed in recent decades have unexpectedly shown that, contrary to what was originally hypothesized, numerous single-target drugs behave as multiple ligands in vivo [6]. In clinical practice, it is not easy to predict the in-vivo effects of a molecule capable of acting on different targets. In fact, depending on the dose and its pharmacokinetics, it could generate both positive or negative effects especially if used in treatment of chronic multifactorial diseases. However, the belief that MDLs can offer many advantages over combinatorial mono-therapies has prompted many researchers in developing new multiple-ligands drugs for T2D treatment (Figure 3). In principle, a molecule capable of acting as an MDL should offer the advantages of a combination therapy but with fewer side effects.

Although this idea is considered exciting, the identification of appropriate multifunctional scaffolds represents the main challenge for researchers engaged in the generation of new MDLs. The debate among scientists regarding the best strategy to obtain the best multifunctional scaffold remains heated. Large-scale screening and knowledge-based approaches are considered the most effective strategies for designing and developing multi-target molecules. The first approach relies on the fact that a known drug actually behaves like a multiple ligand. Therefore, based on this hypothesis, synthesizing new molecules would no longer be required, but rather it would be sufficient to identify new potential ligands in addition to those already known. Conversely, the knowledge-based approach exploits previous information obtained through structure-activity analysis (SAR) focusing on the mode of interaction between molecules of interest and their biological targets. Then, once more appropriate molecules are identified, these are chemically linked or combined together to generate new multi-target drugs that will be subsequently tested to evaluate their real effectiveness.

Besides those aforementioned, several in-silico methods can be used to select potential pharmacophores useful for assembling new MDLs. The advantage of using the computational approach is its ability to easily perform high-throughput analyses, starting from databases containing hundreds to thousands of different compounds. The selected pharmacophores then can be linked together to produce new MDLs. In a forward step, the most promising molecules can be further modified to improve their affinity towards the ligands and safety profile while reducing their toxicity [7].

### 1.3. New MDLs for Treatment of T2D

The emerging interest of scientists and pharmaceutical companies versus multiple ligand drugs is evidenced by the growing number of MDLs produced in the last years for treatment of T2D. Coskun and co-workers projected and synthesized dual glucose-dependent insulinotropic polypeptide receptor (GIP-R) and glucagon-like peptide-1 receptor (GLP-1) agonists and demonstrated that treatment with such molecule stimulated insulin release, leading to a significant reduction of both fasting and postprandial glycaemia [8]. In recent years, convincing evidence suggested that GLP-1 acts as anorexigenic peptide binding to GLP-1R in the hypothalamic region, inducing satiety [9]. Interestingly, it has been demonstrated that, in the brain, GLP-1 acts synergistically with PYY (peptide YY), a peptide that is co-secreted with GLP-1 from enteroendocrine L cells of the intestine and binds NPY2R (Neuropeptide Y receptor Y2 receptor). This finding stimulated many researchers to evaluate the activity of new GLP-1R/NPY2R agonists as antidiabetic agents. It has been demonstrated that GLP-1R/NPY2R dual agonists in vivo exert anorectic effects along with the ability to reduce blood glucose levels, thereby confirming that they could act as promising anti-obesity and antihyperglycemic agents [10].

Among all biological targets, peroxisome proliferation-activated receptors (PPARs) are considered some of the most effective ones for treatment of T2D. Various type of PPARs are differently expressed in human tissues; namely, PPAR-α, δ, and γ, have been identified and characterized to date. The PPAR-α is highly expressed in liver, kidney, heart muscle, and vascular endothelial cells, where its activation promotes fatty acid oxidation, thereby avoiding accumulation of intracellular lipids depots. Treatment with PPAR-α agonists increases cardiac performances in diabetic patients, reducing the risk of stroke [11]. PPAR-δ is ubiquitously expressed, and its activation leads to different effects. In muscle cells, PPAR-δ activation stimulates fatty acids oxidation, while it reduces glucose utilization. In adipose cells, PPAR-δ activation increases the expression of genes involved in fatty acids β-oxidation and energy dissipation via uncoupling of fatty acids oxidation and ATP production [12].

Interestingly, it has been demonstrated that the balance of fatty acids oxidation and synthesis can affect inflammatory and immunosuppressive T cells and macrophages. In macrophages, PPAR-δ activation impairs polarization toward M2-like phenotype with reduced inflammatory potential. According to this hypothesis, antidiabetic functions of PPAR-δ have been associated with reduced inflammatory signalling [13]. PPARγ is largely expressed in adipose tissue, and its activation promotes proliferation and differentiation of preadipocytes into adipocytes. Moreover, PPARγ agonists stimulate deposition of fatty acids into adipocytes, lowering fatty acids blood levels, preventing hyperlipidemia, and increasing peripheral insulin sensitivity [14]. Saroglitazar, a PPARα/γ dual agonist, has been recently approved in India for treatment of diabetic dyslipidemia based on the promising results obtained in clinical trials. Diabetic patients treated with Saroglitazar showed no adverse events, an improved lipidemic profile, an increased insulin sensitivity and β-cell function [15]. Recently, a novel dual peroxisome proliferator-activated receptor alpha/delta (PPAR-α/δ) agonist was synthesized and tested on animal models. Besides protecting the liver from inflammation, and fibrosis, the administration of dual agonists decreased hepatic lipids accumulation, protecting animals from the development of liver steatosis [16,17]. Finally, many efforts have been made to generate pan PPAR agonists combining the pharmacophore motif of PPAR-α, β, and ɣ agonists. Such molecules reduce lipids accumulation in the liver and improve liver damage, inflammation, fibrosis, and insulin resistance [18,19]. Although many new pan PPAR agonists have demonstrated their efficacy as antidiabetic drugs in the preclinical phase, subsequent clinical studies have shown their limitations and revealed their intrinsic toxicity. For these reasons, studies on these molecules have not progressed further [20].

Very recently, Qi Pan and coworkers evaluated the antidiabetic activity of GLP-1-Fc-FGF21 on diabetic and obese mice models. This new dual targeting agonist, able to target both the GLP-1 and FGF21 (Fibroblast growth factor 2) pathway, showed a potent antihyperglycemic activity and caused a marked weight loss, suppressing the appetite and reducing caloric intake. Together, these results suggested that GLP-1/FGF21 dual agonists possess all characteristics to become promising new drugs to fight diabetes and obesity [21].

### 1.4. Dual α-Glucosidase/PTP1B Inhibitors: A New Drugs against Type 2 Diabetes?

Clinical studies revealed that many people show evident signs of metabolic abnormalities years before the diagnosis of T2D. Insulin resistance (IR) is one of the most common abnormalities. IR can affect liver, skeletal muscle, adipose tissue, pancreas, and hypothalamic region, generating several metabolic dysfunctions and promoting the onset of T2D. Besides genetic factors, a diet rich in simple carbohydrates, sedentary behavior, and obesity are thought to be the main risk factors responsible for the development of insulin resistance [22]. These evidence suggest that all measures that limit glucose absorption and increase insulin sensitivity should be the first-line approaches recommended to reduce the risk of developing T2D.

We are convinced that MDLs targeting both PTP1B and α-glucosidases could be used to reach the goal (Figure 4).

The tyrosine phosphatase 1B (PTP1B) acts as a key negative regulator of insulin receptor, and a plethora of studies confirmed that uncontrolled activity of this enzyme is one of the main causes that lead to IR [23]. According to this hypothesis, it has been demonstrated that the overexpression of PTP1B promotes IR in liver [24], muscle [25], adipose tissue [26], pancreas [27], and brain [28]. Conversely, many studies confirmed that PTP1B downregulation or inhibition improves insulin sensitivity, normalizes blood glucose levels, and protects from obesity and the onset of T2D [29]. Overall, such evidence suggest that PTP1B targeting could generate a pleiotropic effect, improving insulin response in liver, muscle adipose tissue, pancreas, and brain, thereby correcting most metabolic abnormalities observed in diabetic patients. Since PTP1B does not have a role in regulating intestinal absorbance of glucose, it is improbable that small molecules designed to target this enzyme could be used to regulate intestinal absorption of glucose.

Monosaccharides, such as glucose, fructose, and galactose, are the only sugars absorbed by gut. Oligosaccharides derived from starch digestion are processed by pancreatic α-amylase and intestinal α-glucosidase to produce free glucose that is then uploaded from intestinal cells. Therefore, the rate of blood glucose raising mainly depends on the gut glucose concentration that, in turn, is influenced by the activity of glucosidases present in the gut. This finding inspired many researchers to challenge glucosidase inhibitors as pharmaceutical tools for the treatment of T2D based on the hypothesis that such molecules could delay the release of glucose from complex carbohydrates, slowing down the rise in blood sugar levels observed after a meal. In the last decades, different kinds of glucosidases inhibitors have been produced and approved as antihyperglycemic drugs [30]. Today, such molecules are used as first-line therapy for T2D patients or administrated in combination with other oral anti-diabetics drugs when metformin/biguanides mono-drug based therapies failed the achievement of the glycemic goal [31]. The evidence that glucosidases inhibitors act synergistically with different oral antihyperglycemic drugs suggested that α-glucosidase/PTP1B dual inhibitors could be successfully projected and used as drugs for treatment of T2D.

### 1.5. Synthesizing New α-Glucosidase/PTP1B Inhibitors

Some studies conducted in the last three years demonstrated the potential effectiveness of dual α-glucosidase and PTP1B inhibitors. In 2017, Mei-Yan Wang et al. demonstrated that (azole-2-yl)-sulfonylalkanamides can target both α-glucosidase and PTP1B, paving the way for the development of new MDL antidiabetic drugs. The most potent compound among them showed IC_50_ values for α-glucosidase and PTP1B of 10.96 and 13.45 µM, respectively, and a good selectivity for PTP1B [32]. Two years ago, Xhenti Ferhati et al., demonstrated that by linking an iminosugar moiety with a phosphotyrosine mimetic, it is possible to create a new generation of antidiabetic drugs affecting both PTP1B and α-glucosidases and showing IC_50_ values for α-glucosidase and PTP1B in the 4–200 µM range. Moreover, tests carried out on HepG2 cells demonstrated that some of these compounds show a good insulin-mimetic activity, enhancing phosphorylation levels of Akt in the absence of insulin stimulation. We can hypothesize that, in absence of insulin, the PTP1B inhibition results in an enhancement of insulin receptor phosphorylation level, promoting the activation of insulin signaling pathway [33].

Finally, in 2020, Malose J. Mphahlele et al. investigated the properties of a series of ortho-hydroxyacetyl-substituted 2-arylbenzofuran derivatives, showing that some of these have IC_50_ values in the submicromolar and in the micromolar range for α-glucosidase and PTP1B, respectively [34].

## 2. Nature-Inspired Scaffold Molecules for the Synthesis of Dual α-Glucosidase/PTP1B Inhibitors

It is well known that natural sources, such as plants, fruits, algae, and microorganisms, are important sources of bioactive molecules that often have been used as lead compounds to the development of new drugs for treatment of human diseases [35]. Based on this evidence, we analyzed literature data looking for natural molecules showing both α-glucosidase and PTP1B inhibitory activity that should be used as scaffold molecules for the synthesis of new dual-target antidiabetic drugs. Data collection from the literature was performed by querying the PUBMED database, using specific keywords, such as “PTP1B and alpha-glucosidase inhibitors” (which yielded 79 results), “alpha-glucosidase and PTP1B” (48 results), “Dual targeting PTP1B and glucosidase” (6 results), and “PTP1B and multitarget inhibitors” (6 results). Every single study was downloaded and analyzed in depth to extract the data of interest. The collected compounds were classified into twelve different groups based on their chemical structure. Surprisingly, we found that, in the last twenty years, more than 200 compounds showing dual α-glucosidase/PTP1B inhibitory activity have been discovered and characterized. To make it easier for the reader to analyze the data, we have divided the identified compounds into different classes.

### 2.1. Coumarins

Sixteen coumarin-derivatives were isolated from *Angelica decursiva* (compound **1**–**7**, **12**, **13**, and **17**), from *Artemisia capillaris* (compounds **9**–**11**, **14**, and **16**), and from *Euonymus alatus* (Thunb.) Sieb (compound **8**) (Table 1).

The dihydroxanthyletin-type family includes (+)-*trans*-decursinol (**1**), Pd-C-I (**2**), Pd-C-II (**3**), Pd-C-III (**4**), 4′-Hydroxy Pd-C-III (**5**), and 4′-Methoxy Pd-C-I (**7**). Among these, (+)-*trans*-decursinol (**1**), a molecule bearing two hydroxyl groups on C3 and C4 of pyrene moiety, resulted the most potent inhibitor of both targets, showing IC_50_ values of 2.33 and 11.32 µM for PTP1B and α-glucosidase, respectively. Conversely, Decursinol (**12**), which is lacking in OH on C4, showed a lower affinity for both targets. This result suggested the pivotal role of this hydroxyl group in improving the inhibitory activity of natural dihydroxanthyletin-type coumarins. Interestingly, the replacement of OH on C4 with a senecioyl group produced Pd-C-II (**3**), which exhibited a slightly decreased inhibitory activity in respect to (**1**), indicating that this group only partially succeeds in substituting the hydroxyl group. Pd-C-I (**2**), a derivative bearing a senecioyl group on C3, showed IC_50_ values for PTP1B and α-glucosidase similar to those of (+)-*trans*-decursinol. On the other hand, 4′-Hydroxy Pd-C-III (**5**), bearing an angeloyl group on C3, maintained a significant inhibitory activity toward PTP1B but showed a weaker inhibitory activity on α-glucosidase in comparison to (+)-*trans*-decursinol, suggesting that the nature of aliphatic chain is crucial to stabilize the α-glucosidase-inhibitor complex. The substitution of 4 OH with an acetyl group leads to Pd-C-III (**6**), which showed a lower affinity for PTP1B but a better inhibitory capacity towards the α-glucosidase compared to (**5**). Conversely, the insertion of a methoxy group on C4 of Pd-C-I leads to Methoxy Pd-C-I (**7**), a compound showing an inhibitory activity on PTP1B comparable to that of parental molecule but a slower affinity for the α-glucosidase. Finally, Decursidin (**12**), bearing two senecioyl groups on C3 and C4, respectively, showed a reduced inhibitory activity for both targets compared to (+)-*trans*-decursinol, thereby confirming the OH groups are important to reinforce the binding of dihydroxanthyletin-type coumarins with PTP1B and α-glucosidase. Among phenyl-coumarins, Selaginolide A (**15**), a 7-hydroxycoumarin derivatives bearing a 4-hydroxy-3,6-dimethoxyphenyl ring linked to C3, was proven the most potent compared to Euonymalatus (**8**), showing the latter had a more complex structure and several OH groups. However, both compounds showed a balanced and potent inhibitory activity for both targets, suggesting that phenolic coumarins are good lead compounds to generate new antidiabetic drugs acting on both enzymes. Umbelliferone (**16**) or 7-hydroxycoumarin moiety is a prototype of several dual PTP1B and α-glucosidase inhibitors, such as Esculetin (**9**), Daphnetin (**8**), Scopoletin (**14**), Umbelliferone 6-carboxylic (**17**), and 6-Methoxy artemicapin C (**10**). Daphnetin, bearing an additional OH group on C8 compared to Umbelliferone, showed a slight increased inhibitory activity for both targets, while Esculetin, a 6,7 dihydroxy coumarin, showed IC_50_ values for PTP1B and α-glucosidase 27 and 7 times lower than those calculated for Umbelliferone. Interestingly, Scopoletin, bearing a methoxy group on C6, showed an affinity for PTP1B similar to that of Umbelliferone but an increased affinity for α-glucosidase in comparison to them. Conversely, Umbelliferone 6-carboxylic showed high affinity for PTP1B (IC_50_ = 7.98 µM) but a weak affinity for α-glucosidase (IC_50_ = 172.10 µM). Finally, the mono-hydroxylated compound 6-Methoxy artemicapin C (10), showing a methoxy group on C7, had a significant inhibitory activity for PTP1B (IC_50_ = 27.6 µM) but a weak inhibitory power for α-glucosidase (IC_50_ = 563.7 µM). This data suggested that substituents on C6 can be relevant to potentiate the interaction of inhibitor to both targets even if OH and negative-charged groups on C6 favor the interaction with PTP1B, while non-polar groups in this position favor the interaction with α-glucosidase.

### 2.2. Lignans

Numerous lignans showing inhibitory activity toward both PTP1B and α-glucosidase have been recently isolated from various plants. Compounds **18**–**24** and **41**–**48** have been isolated from *Viburnum macrocephalum* f*. keteleeri* [41]. Moreover, compounds **25**–**31** and **33**–**40** were extracted from the roots of *Limonium gmelinii* (Willd.) Kuntze [42], while compound **32** was obtained from *Hizikia fusiformis* (Harvey) Okamura [43], an edible marine alga, while (Table 2).

Lignans glycosides family accounts several members with chemical differences that give them peculiar properties. The (+)-pinoresinol 4-*O*-*β*-d-glucoside (**22**) and (+)-syringaresinol 4-*O*-*β*-d-glucoside (**23**) showed high IC_50_ values for both enzymes, suggesting that pinoresinol scaffolds bearing vanilloyl or syringoyl units, per se, did not guarantee a tight binding to both targets. Interestingly, the introduction of an OH group on C8 (**21**) or C8′ (**20**) strongly increased the affinity for PTP1B but slightly improved the inhibitory power for α-glucosidase. On the other hand, the addition of a *p*-hydroxybenzoyl (**45**) or a vanilloyl group (**46**) to C6″ slightly increased the affinity for PTP1B without improving the affinity for α-glucosidase. Noticeably, the addition of a glucosyl or galactosyl group on C4″ (**41**) or C2″ (**42**), respectively, slightly increased the affinity of such molecules for PTP1B but exhibited significant inhibitory activity for α-glucosidase (IC_50_ = 16.7 and 17.1 µM, respectively). A significant affinity increase for both targets (IC_50_ for PTP1B and α-glucosidase of 25.8 and 16.1 µM, respectively) was obtained introducing a vanilloyl-4-*O*-*β*-glucopyranosyl group on C6″ (**43**), while Viburmacroside G and H, bearing, respectively, a syringoyl group and an α-l-rhamnopyranosyl residue (**47**) or a syringoyl group and α-d-xylopyranosyl unit (**48**), both showed a weaker inhibitory activity compared to (**43**), thereby confirming the key role of chemical substituents linked to C6″ in determining the inhibitory power of molecules. Finally, Viburmacroside D (**44**) bearing an apiofuranosyl group located at C2″″ of the glucopyranosyl moiety, showed the highest inhibitory activity of these series (IC_50_ of 8.9 and 9.9 µM for PTP1B and α-glucosidase, respectively). In conclusion, these data suggest that the inhibitory power of Viburmacrosides in respect to α-glucosidase is attributable to the presence of a hydroxyl group at C4′ and of two or more monosaccharide units at C4. Meanwhile, the ability of such compounds to inhibit PTP1B seem to be influenced by the presence of hydroxyl groups on C8 or C8′, by a glycosyl-phenolic acyl residue located at C6″, and mainly by an apiofuranosyl group located at C2″″ of the glucopyranosyl moiety.

As far as lignanamides derivatives are concerned, these represent a highly structurally heterogenous family whose members showed a different inhibitory activity for both targets. Among these, Cannabisin I (**31**) proved to be the most active compound, showing an IC_50_ value for PTP1B and α-glucosidase of 2.01 and 1.5 µM, respectively. It is interesting to note that Limoniumin B (**34**), bearing a methoxyl group on C3′ instead of one OH, showed an IC_50_ value for α-glucosidase comparable to that of compound (31) and a three-times higher IC_50_ value for PTP1B. However, Limoniumin C (**35**), bearing a methoxy group on C3, showed similar affinity for PTP1B to compound (**35**) but one order of magnitude lower inhibitory activity for α-glucosidase. Finally, substitution of OH groups on C3 and C3″ led to Limoniumin D (**36**), a compound acting as a weaker inhibitor than (**35**). Intriguingly, Limoniumin E (**37**), which lacks the dihydroxybenzene group linked to C7′, showed an increased IC_50_ for α-glucosidase but maintained a significant inhibitory activity for PTP1B.

The role of the OH groups in determining the inhibitory activity of lignanamides is evidenced for compounds showing a phenyldihydronaphthalene core, such as Limoniumin H (**39**), I (**40**), Cannabisin D (**29**), B (**27**), and C (**28**). Cannabisin B, bearing two OH groups on C3 and C3′, showed a high affinity for both targets (IC_50_ values for PTP1B and α-glucosidase were 5.89 and 4.56 µM, respectively), whereas compounds (**28**) and (**39**), bearing respectively a methoxyl group on C3′ and on C3, exhibited higher IC_50_ values for α-glucosidase. Moreover, the introduction of two methoxyl groups (on C3 and C3′) cause a strong loss of inhibitory activity of compound (**29**). It is interesting to note that the affinity of compound (**40**) for α-glucosidase is eight times lower than (**39**), suggesting that the position of OH groups on naphthalene moiety can influence the interaction of lignanamides with this enzyme.

### 2.3. Xanthones

Several alkylated xanthones active on both PTP1B and α-glucosidases have been extracted from roots bark of *C. Cochinchinense* [44] (Table 3).

All compounds isolated behaved as good inhibitors of both targets, showing IC_50_ values in the 1.7–80 µM range for α-glucosidase and between 2.8 and 52.5 µM for PTP1B. Although selected xanthones possess different substituents, they behaved as mixed-type inhibitors of α-glucosidases and competitive inhibitors of PTP1B. This finding suggests that xanthone moiety has a key role in determining the interaction with the active site of both targets even if aliphatic chains and hydroxyl groups linked to xanthone structure can influence the affinity of each compound.

The γ-Mangostin (**49**), bearing two prenyl chains on C2 and C8 and four OH groups, resulted as the most potent inhibitor of this series, showing IC_50_ values for PTP1B and α-glucosidase of 2.8 and 1.7 µM, respectively. The replacement of OH group on C7 with a methoxyl group (α-Mangostin, **50**) slightly affected affinity for both targets, while the affinity of Cratoxylone (**59**), also bearing an hydroxylated prenyl chain on C2, was reduced eight times for PTP1B and eighteen times for α-glucosidase. The displacement of prenyl chain from C8 to C4 impaired the inhibitory activity of 1,3,7-Trihydroxy-2,4-diisoprenylxanthone (**51**), while the replacement of prenyl chain on C4 with a geranyl group (Cochinechinone A, **55**) strongly improved the affinity of the molecule for PTP1B. Interestingly, hydroxylation of the prenyl chain on C2 (Caratoxanthone A, **53**) but not the hydroxylation of the geranyl chain on C4 (Cratoxanthone F, **58**) makes compound (**53**) an inhibitor almost as efficient as compound (**49**). The Pruniflorone S (**60**) showed IC_50_ values similar to those of compound (**55**), suggesting that the substitution of OH groups with an aliphatic chain did not result in a further enhancement of inhibitory power. 7-Geranyloxy-1,3-dihydroxyxanthone (**52**) and Cochinxanthone A (**56**) showed a weak inhibitory activity toward both targets, indicating that the presence of a single geranyl chain was not functional to improve the inhibitory activity of these compounds. Finally, Cochinchinone Q (**54**), which does not bear aliphatic chains, resulted as the weaker inhibitor of this series toward both targets, thereby confirming the importance of aliphatic chains in stabilizing the xanthones-enzyme complexes.

### 2.4. Terpenes

Many terpenes acting on both α-glucosidase and PTP1B have been isolated in the last years from *Aralia elata* (**62**, **63**), *Euonymus alatus* (Thunb.) Sieb. (**71**, **74**), *Hizikia fusiformis* (**64**, **65**), *Agrimonia pilosa* (**80**, **81**), *Panax ginseng* C.A. Meyer (**66**–**70**), *Myrtus communis* Linn. (**72**, **73**, **75**, **78**, **79**), and *Pueraria lobata* (**76**, **77**) (Table 4).

Among all molecules belonging to this family, two glycosylated terpenes, mainly 3-*O*-[*β*-d-Glucopyranosyl (1→3)-*β*-d-glucopyranosyl]-caulophyllogenin 28-*O*-*β*-d-glucopyranosyl ester (**62**) and 3-*O*-*α*-l-Arabinopyranosyl echinocystic acid (**63**), isolated from the leaves of *Aralia elata*, proved to be potent α-glucosidase inhibitors (IC_50_ value of 0.73 µM and 2.96 µM, respectively). Although their action mechanism is not known, it is probable that glycosyl groups mimic substrate, favoring the access of molecules into active site of enzyme. However, non-glycosylated terpenes can also act as potent α-glucosidase inhibitors. Recently, five ginsenosides, 20(*R*)-25-Methoxydammarane-3*β*,12*β*, 20-tetrol (**66**), 20(*R*)-Dammarane-3*β*,6*α*,12*β* 20, 25-pentol (**67**), 20(*R*)-Protopanaxadiol (**68**), 20(*R*)-Protopanaxatriol (**69**), and 20(*S*)-Panaxatriol (**70**), showing different inhibitory power toward α-glucosidase, have been isolated. Interestingly, the affinity for α-glucosidase decrease in the order (**67**) > (**66**) > (**69**) > (**70**) > (**68**), suggesting that the number and position of hydroxyl groups of molecules are essential to stabilize the protein-ligand interaction, probably through formation of an extensive hydrogen bond network with atoms of the enzyme. In agreement with this hypothesis, 24*R*-Methyllophenol (**71**) and Epi-lupeol (**75**), two molecules bearing a single hydroxyl group, behaved as weaker α-glucosidase inhibitors, showing IC_50_ values of 183.1 and 133.1 µM, respectively. Among pentacyclic triterpenes (**72**, **73**, **75**, **78**, and **79**), Tormentic acid, bearing a carboxyl group on C5, and three hydroxyl groups (on C1, C20, and C21) resulted as the best inhibitors, with an IC_50_ value of 23.8 µM for α-glucosidase. On the other hand, Oleanolic acid (**79**), bearing a carboxyl group on C17 and a single hydroxyl group on C1, showed a similar IC_50_ value (34.3 µM). Moreover, displacement of a methyl group from C20 to C19 (Ursolic acids, 81, IC_50_ = 42.4 µM) and the insertion of an additional hydroxyl group on C2 (Corosolic acid, 73, IC_50_ = 55.5 µM, and Maslinic acid, **78** IC_50_ > 100 µM) caused loss of the inhibitory activity. These results also indicated that the methyl group on C20 of pentacyclic triterpenoids might be required for α-glucosidase-inhibitory activity. Finally, the insertion of additional groups on C19 (Betulinic acid, **72**, IC_50_ > 100 µM), the removal of carboxyl group on C17 (Lupeol, **77**, IC_50_ = 176 µM), the insertion of a chemical substituent on C1 (Jacoumaric acid, **75**, IC_50_ > 100), as well as the displacement of carboxyl group on C20 (18*α*/*β*-Glycyrrhetinic acid, **64** and **65**, IC_50_ = 113.30/128.7 µM for α-glucosidase) caused a dramatic loss of the activity of such molecules for this target. As far as the inhibitory activity on PTP1B is concerned, all compounds behave as good inhibitors, showing IC_50_ values in the 0.5–38.9 µM range. Among all, the most potent PTP1B inhibitors were Tormentic acid (IC_50_ 0.5 µM), Ursolic acid (IC_50_ = 3.4 µM), Oleanolic acid (IC_50_ = 8.9 µM), and 20(*R*)-Dammarane-*3β*,6*α*,12*β* 20, 25-pentol (IC_50_ = 10 µM). Of note, compounds **62**, **63**, and **66**–**82** showed a good inhibitory activity against both targets, suggesting that terpenic moiety represents a versatile scaffold structure to generate new bifunctional inhibitors targeting both α-glucosidase and PTP1B.

### 2.5. Glycosides

Several glycosylated molecules active on both PTP1B and α-glucosidase were isolated from different sources, such as *Euonymus alatus* (Thunb.) Sieb. (**82**), *Agrimonia pilosa* Ledeb. (Rosaceae) (**85**–**87**), *Pueraria lobata* (**83**, **88**, **89**), *Citrus reticulata* (**84**), and *Myrcia* rubella (**90**, **91**) (Table 5).

Among glycosides, Quercetin-3-(2-*E*-sinapoyl)-*O*-glucopyranoside (**90**) and Quercetin-3-*O*-(6′′-*O*-E-*p*-feruloyl)-*β*-d-glucopyranoside (**91**) were the most active, showing IC_50_ values on α-glucosidase and PTP1B of 4.96 and 10.86 µM and 64.9 and 39.4 µM, respectively. Interestingly, Quercetin-3-*O*-*β*-d-glycoside (**87**) and Kaempferol-3-*O*-α-l-rhamnoside (**85**) possessed a good inhibitory activity toward both targets, showing the first IC_50_ values of 27.6 and 29.6 µM and the second of 12.16 and 28.5 µM for PTP1B and α-glucosidase, respectively. However, Apigenin-7-*O*-*β*-d-glucuronide-6′′-methyl ester (**86**) maintained a strong inhibitory activity toward PTP1B but exhibited a weaker activity for α-glucosidase, thereby confirming that glucosyl moiety has an important role in the targeting α-glucosidase. These results suggest that glycosylate flavonoids, per se, are potent inhibitors of both targets and suitable scaffold molecules for synthesis of new MLDs. The fact that 1-Feruloyl-*β*-d-glucoside had a marginal role in determining the inhibitory power of (**90**) and (**91**) is supported also by evidence that this group alone (**82**) behaved as weak α-glucosidase inhibitors (IC_50_ = 169.7 µM) but showed a relevant inhibitory activity toward PTP1B (IC_50_ 18.4 µM). Similarly, the relevant activity of Didymin (**84**) for both targets (IC_50_ = 1.2 and 48.7 µM for PTP1B α-glucosidase, respectively) suggests that methoxyflavanone moiety can be used to generate potent PTP1B and α-glucosidase inhibitors. Conversely, isoflavone moiety seems not to be as good a scaffold as glycosyloxyisoflavones, such as Daidzin (**83**) and Puearin (**88**) showed a reduced affinity toward both targets. Finally, the substitution of isoflavone moiety of Puerarin with a different scaffold (Puerarol B-2-*O*-glucoside, **89**) caused a further loss of the inhibitory activity for both targets, confirming that polyphenol structure played an important role in PTP1B and α-glucosidase inhibition.

### 2.6. Fatty Acids

Studies conducted on extracts obtained from *Hizikia fusiformis* (Harvey) Okamura lead to identification of several fatty acids (**92**, **93**, **96**–**98**) as natural α-glucosidase and PTP1B inhibitors. Further fatty acids active on both targets (**94**, **95**) were obtained from *Agrimonia pilosa* Ledeb. (Rosaceae) (Table 6).

The isolated compounds (*Z*)-Hexadec-12-enoic acid (**92**), Palmitic acid (**95**), (7*Z*,9*Z*,11*Z*,13*Z*)-Eicosa-7,9,11,13-tetraenoic acid (**97**), and (8*Z*,11*Z*,14*Z*)-Heptadeca-8,11,14-trienoic acid (**98**) exhibited the strongest inhibitory activity, with IC_50_ values comprised between 0.5–11.5 µM for PTP1B and between 34.8–48.05 µM range for α-glucosidase. Overall, Palmitic acid, a saturated fatty acid, seemed the most potent inhibitor of PTP1B, showingshowing an affinity for this enzyme similar to that of some unsaturated fatty acids. The only exception is Methyl 2-hydroxyl tricosanoate (**94**), which showed a reduced affinity for the enzyme. This data suggest that methylation of carboxylic acid has a detrimental effect in terms of inhibitory power and that carboxylic acid group contributes to stabilize the enzyme-inhibitor complex, probably forming hydrogen bonds or ionic interactions with some amino acids of the target enzyme. Furthermore, as far as the inhibitory activity on α-glucosidase is concerned, is not possible to define a strict correlation between fatty acids structure, number of carbon atoms, and inhibitory activity. Palmitic acid, a saturated fatty acid, showed similar activity (IC_50_ = 45.5 µM) to (*Z*)-Hexadec-12-enoic acid (**92**, IC_50_ = 48.05 µM), a monounsaturated fatty acid or that of (8*Z*,11*Z*,14*Z*)-Heptadeca-8,11,14-trienoic acid (**98**, IC_50_ = 43.9 µM), a heptadecenoic acid having three double bonds located at positions 8, 11, and 14. In addition, (*Z*)-Octaec-9-enoic acid (**93**) and (7*Z*, 10*Z*, 13*Z*)-Octadeca-7,10,13-trienoic acid (**96**), a monounsaturated fatty acid the former and a polyunsaturated fatty acid the second, showed a similar, never unexpectedly weak inhibitory activity towards α-glucosidase (IC_50_ values = 113 and 111 µM, respectively).

### 2.7. Anthraquinones

Anthraquinones reported in Table 7 were obtained from *Cassia obtusifolia* L., a legomnous annual herb growing in tropical countries of Asia [44] (Table 7).

Screening assays carried out using both PTP1B and α-glucosidase revealed that Alaternin (100) is the most active compound, showing an IC_50_ value in the low micromolar range (IC_50_ = 1.22 and 0.99 µM for PTP1B and α-glucosidase, respectively). Furthermore, kinetic analyses revealed that Alaternin acts as a competitive inhibitor of PTP1B and a mixed type inhibitor of α-glucosidase. In addition, docking analyses performed with PTP1B showed that hydroxyl groups present on C1, C2, and C6, as well as methyl group present on C3 of Alaternin, have a key role in stabilizing the Alaternin-PTP1B complex. Moreover, although no kinetic and docking data are available on α-glucosidase, it is reasonable to think that hydroxyl groups contribute to stabilize the complex Alaternin/α-glucosidase, too. According with this hypothesis, we found that the introduction of a methoxyl group on C1 of Alaternin generates 2-Hydroxyemodin-1 methylether (**99**) that showed a reduced activity for both targets. Moreover, the removal of the OH group on C6 leads to Obstusifolin (**113**), a molecule showing a good affinity for PTP1B but a reduced affinity for α-glucosidase. On the other hand, the removal of the OH group from C2 leads to Emodin (**110**), a molecule that possesses an IC_50_ value unchanged compared to Alaternin but with a lower affinity for PTP1B. In addition, Chrysophanol (**107**), obtained by removing the OH group from C6 of Emodin, showed a very weak affinity for α-glucosidase (IC_50_ = 0.99 µM of Alaternin versus 46.81 µM of Chrysophanol) and a moderate decrease of the affinity for PTP1B (1.22 µM for Alaternin versus 5.86 µM of Chrysophanol). The replacement of the methyl group on C3 with an alcoholic group leads to Aloe-emodin (**101**), which showed a reduced affinity for PTP1B but behaved as a potent α-glucosidase inhibitor (IC_50_ = 1.4 µM). On the other hand, the introduction of a methoxyl group on C8 of Emodin leads to Questin (**116**), which behaved as a weaker α-glucosidase inhibitor than Emodin, whereas the presence of a methoxyl group on C3 generates Physcion (**115**), a molecule that maintained a relevant affinity for both targets. Furthermore, the introduction of a methoxy group on C1 leads to 2-Hydroxyemodin-1 methylether (**99**), a molecule showing higher IC_50_ values for both targets, while the removal of OH group from (**99**) generates Obstusifolin (**113**), showing a good affinity for PTP1B but a poor affinity for α-glucosidase (IC_50_ = 142.12 µM). Finally, the introduction of an additional methoxy group on C7 of (**99**) leads to Aurantio-obtusin (**102**), which presented IC_50_ values slightly increased in respect to (**99**). It is noteworthy that Obtusin (**114**) and Chryso-obtusin (**105**), showing methoxy groups on C1, C6, and C7 and C1, C6, C7, and C8, respectively, showed IC_50_ values for PTP1B and α-glucosidase lower than those of (113), suggesting that a methoxy group can at least partially replace the function of the OH group. Glycosylated derivatives, such as Chryso-obtusin-2-glucoside (**106**), Chrysophanol tetraglucoside (**108**), and Chrysophanol triglucoside (**109**), showed low affinity for both targets regardless of the position and number of glycosyl groups. Similarly, glycosides bearing a naphthopyrone group, such as Cassiaside (**103**) and Cassitoroside (**104**), behaved as weak inhibitors of PTP1B and α-glucosidase, suggesting that anthraquinone by itself represents a good scaffold for developing new MDLs for the treatment of T2D.

### 2.8. Phenolic Compounds

The family of phenolic compounds showing inhibitory activity toward PTP1B and α-glucosidase includes several and heterogenous molecules extracted from different natural sources, including *Artemisia capillaris* (**135**, **145**, and **175**), *Selaginella rolandi-principis* (**121**), *Pueraria lobata* (**130**, **137**, **138**), *Paeonia lactiflora* (**117**, **134**, **136**, **174**, **176**), *Morus alba* var. tatarica (**119**, **124**, **125**, **160**), *M. alba* L. (Moraceae) (**120**, **126**, **127**, **159**, **161**–**169**, **170**–**173**), *Ecklonia stolonifera* and *Eisenia bicyclis* (**122**, **139**–**141**), *Alpinia katsumadai* (**118**, **128**, **129**, **131**–**133**, **143**, **144**, **146**–**157**), *Amomum tsao-ko* (**175**–**188**), *Paulownia tomentosa* (**142**, **190**–**195**), and *Paeonia delavayi* (**123**, **158**) (Table 8).

The high chemical variability and the elevated number of hydroxyl groups present on such molecules make them interesting scaffold molecules for the development of new MDLs. Flavonoids, such as 6,7-dimethoxy-2′,4′-dihydroxyisoflavone (**121**), Calycosin (**130**), Daidzein (**138**), Eriodictyol (**142**) Isorhamnetin (**145**), Luteolin (**158**), Morin (**165**), and Quercetin (**175**), are good inhibitors of both targets, showing IC_50_ values for PTP1B in the 20–136 µM range and those for α-glucosidase between 4 and 94 µM. Among these, Morin, a flavanol bearing five OH groups, resulted as the most potent inhibitor of both targets, showing IC_50_ values for PTP1B and α-glucosidase of **33** and 4.48 µM. Of note, flavanols, such as Quercetin, which differs from Morin in the position of the OH groups on the B ring, and Isorhamnetin, which carries a methoxy group on C3′, showed a slight decrease in affinity towards PTP1B and IC_50_ values 13- and 31-times higher, respectively, for α-glucosidase. These data suggest that the number and the position of the OH groups are key factors in determining the inhibitory power of flavonols mainly for α-glucosidase. Furthermore, Luteolin and Eriodictyol, a flavone and a flavanone that, like Quercetin, bear two OH groups on C3′ and C4′, showed higher IC_50_ value for PTP1B (IC_50_ = 64 and 136 µM, respectively) than Quercetin. Surprisingly, Eriodictyol (IC_50_ value of 10.4 µM) behaved as a better inhibitor of α-glucosidase than Quercetin and Luteolin, suggesting that the presence of a pyran-4-one moiety improves the affinity of flavonoids for α-glucosidase. Among the isoflavones, Calycosin (**130**) and 6,7-dimethoxy-2′,4′-dihydroxyisoflavone (**121**), both bearing an OH group on B-ring and on C7, showed a stronger inhibitory activity for α-glugosidase than PTP1B, while 6,7-dimethoxy-2′,4′-dihydroxyisoflavone showed a higher affinity for PTP1B in respect to α-glucosidase. This finding suggested that, by changing position and number of OH and methoxyl groups, it is possible to generate inhibitors with a different affinity for both targets. Among all compounds, Ugonins appear of particular interest because of their low molecular weight and high inhibitory activity. Ugonin J (**190**) e, 2-(3,4-dihydroxyphenyl)-6-((2,2-dimethyl-6-methylene cyclohexyl)methyl)-5,7-dihydroxychroman-4-one (**191**), was found to be the most effective inhibitor of both PTP1B (IC_50_ = 0.6 and 2.1 µM, respectively) and α-glucosidase (IC_50_ = 3.9 and 7.4 µM, respectively), acting as competitive inhibitors against the first target and as non-competitive inhibitors against the second. Moreover, Ugonin M (**194**), bearing a furane moiety and a cyclohexyl group linked in a different position in respect to (**190**) and (**191**), showed a similar inhibitory activity versus both targets even if acting as a mixed-type inhibitor (binds both free enzyme and enzyme-substrate complex). Finally, Ugonin S (**192**), L (**193**), and U (**195**), bearing a tetrahydro-7*H*-pyran [2,3-c] xanthen-1-one group, showed a slightly weaker affinity for PTP1B but 5–8-times lower inhibitory activity for α-glucosidase compared to Ugonin J. Among high-molecular-weight polyphenols, the most active compounds were 7-Phloroeckol (**122**), Albanol B (**126**), Albasin B (**127**), Dieckol (**139**), Morusalbin A–D (**166**–**169**), Mulberrofuran G (**170**), Mulberrofuran K (**172**), and Yunanensin A (**189**). Interestingly, all these compounds are characterized by a balanced inhibitory activity and high affinity for both targets, as confirmed by the fact that IC_50_ values for both targets fall in the low micromolar range. Interestingly, kinetic and insilico docking analyses revealed that Albasin B, Morusalbin D, and Yunanensin A, some the most active compounds, behaved as mixed-type inhibitors of PTP1B and as competitive inhibitors of α-glucosidase [54].

### 2.9. Prenylated Phenolic Compounds

*Morus alba* L. (**196**, **202**, **206**, **208**, **209**, **210**, **212**), *Paulownia tomentosa* (**197**–**201**, **204**, **205**, **211**), *Glycyrrhiza uralensis* (licorice) (**213**, **214**) are the natural source of prenylated phenolic compounds, reported in the Table 9. 

Such molecules are highly heterogeneous and showed a variable inhibitory activity. Starting from smaller molecules, we observed that, despite their similar structure, Morachalcone A (**212**) is a weaker inhibitor of (*E*)-4-isopentenyl-3,5,2″,4″-tetrahydroxystilbene (**196**), suggesting that the position of dihydroxyphenyl groups influence the activity of such prenylated molecules. Concerning the flavonoid compounds, we observed that Mimulone (**211**), bearing a geranylated-naringenin based structure, is a potent PTP1B inhibitor (IC_50_ = 1.9 µM), and a good α-glucosidase inhibitor (IC_50_ = 30 µM), thereby confirming that flavonoids are good lead molecules for synthesis of new drugs targeting both enzymes. Modifications of Mimulone structure have a different impact on the inhibitory power of chemical derivatives depending on the chemical groups introduced. For instance, the introduction of a methoxy group on C3′ (**199**) or C4′ (**201**) of “B” aromatic ring reduced affinity for PTP1B but improved that for α-glucosidase, whereas the introduction of a second methoxy group on C5′ (**197**) strongly reduced the affinity of the molecule for α-glucosidase. Moreover, the introduction of OH group on C3 of keto-pyrene moiety (**198**, **200**) did not improve the inhibitory power, whereas the presence of two OH groups on C3′and C5′ together with a methoxy group on C4′ generated two molecules, (**204**) and (**205**), showing IC_50_ values in the low micromolar range toward both targets. This finding suggests that the OH group on “B” ring has a key role in enhancing the targets–ligand interactions. Prenylated Morin derivatives, such as Albanin A (**206**), showed a better affinity for α-glucosidase than for PTP1B, while Kuwanon C (**208**), showing two prenyl groups, had increased affinity for both enzymes. A stronger inhibitory activity was observed when prenyl chains were linked to C6 and C8 of Genistein (**203**), suggesting that position of “B” ring strongly influences the inhibitory activity of prenylated phenol molecules. Glycyuralin H (**207**), which has the same 3-hydroxyisoflavanone skeleton of Genistein, behaved as weaker inhibitor of Kuwanon C (**208**), suggesting that modification of “B” ring of Genistein is not a good strategy to improve the inhibitory activity. The presence of a single geranyl chain on C5′ of “B” ring of Morin suitably increased the inhibitory power of 5′-Geranyl-5,7,2″,4″-tetrahydroxy-flavone (**202**) for PTP1B and α-glucosidase. This result indicates that the length of the aliphatic chain can be modulated to improve affinity of inhibitor for both targets. On the other hand, the introduction of additional 2,4-dihydroxyphenyl groups (**209**) resulted in a strong enhancement of inhibitory power of Kuwanon G, thereby confirming that OH groups contribute to strengthening the stability of the enzyme-inhibitor complex. Isoprenylated coumarones proved to be good inhibitors, showing 2′-O-demethylbidwillol B (**213**) similar IC_50_ values for both targets, while Glyurallin A (**214**) appeared to have a greater affinity for α-glucosidase. Finally, the Diels-Alder adduct Macrourin G (**210**) behaved as a potent inhibitor of both enzymes. Interestingly, kinetic and docking analyses revealed that Macrourin G binds into allosteric site identified by Wiesmann in PTP1B [65], while it acts as competitive inhibitor of α-glucosidase, interacting with some residues of catalytic site of enzyme [56].

### 2.10. Caffeoylquinic Acids Derivatives

Several dicaffeoylquinic acid derivatives active on both PTP1B and α-glucosidase were obtain from *Artemisia capillaris* [39] (Table 10).

As the PTP1B inhibitory activity is concerned, all dicaffeoylquinic acids reported in the Table 10 showed significant inhibitory activity. Among these, 1,5 dicaffeoylquinic acid (**215**), the only molecule of this series bearing a caffeic acid molecule linked to C1 of quinic acid, showed the lowest activity overall. Taking into account the differences between the IC_50_ values and chemical structure of compounds reported in Table 10, it is reasonable to hypothesize that the presence of the caffeoyl group at the positions 3, 4, or 5 are related to the higher inhibitory activity of 3,4 and 3,5 caffeoyl derivatives for PTP1B.

Although dicaffeoylquinic acids (**215**–**219**) were also active on α-glucosidase, their inhibitory activity is weaker when compared with that observed on PTP1B. The most potent α-glucosidase inhibitor identified between this group of compounds was the methyl-3,5-di-*O*-caffeoylquinic acid (**219**). Considering the structural differences of (**219**) with 3,5-dicaffeoylquinic acid (**217**), it is possible to infer that the methyl ester bridge at the carboxylic acid is functional to enhance the inhibitory activity against α-glucosidase.

### 2.11. Alkaloids

Eighteen different alkaloids able to inhibit both PTP1B and α-glucosidase were extracted from *Clausena anisum-olens* [66] (Table 11).

It is interestingly to observe that among all carbazole, Clausenanisine A (**224**) exhibited the lowest IC_50_ values on both targets, suggesting that the carbazole moiety bearing a five-membered cyclic ether, a methoxy group, and a short aliphatic chain represents a promising lead structure for developing new MLDs active on both PTP1B and α-glucosidase. Taking into account that Clausenanisine B (**225**) showed inhibitory activity similar to Clausenanisine A (**224**), we argue that the insertion of a tetrahydro-pyran-4-one group resulted in a slight decrease of the affinity for both targets. The loss of the OH group from C2′ of Clausenanisine B (**225**), of both OH and carbonyl groups, or the introduction of a methoxyl or hydroxyl group on C8 of carbazole moiety generated Euchrestifoline (**236**), Dihydromupamine (**235**), Clauraila B (**221**), and Kurryame (**237**), whose affinity for PTP1B and α-glucosidase steadily decreased. However, the most relevant decrease of affinity occurred after the loss of the carbonic group present on the pyrene moiety, suggested that ketocarbonyl group on C1′ of Clausenanisine B is responsible of the significant inhibitory activity of molecule for both enzymes. Clausenanisine F (**229**), which bears a carboxyl group on C3 and an OH group on C1, showed lower but always significant inhibitory activity for PTP1B while it proved to be a very weak inhibitor of α-glucosidase. The addition of a methoxy group on C2 or C6 of (**229**) leads to Clausenanisine C (**226**) and Clausenaline E (**222**), two molecules with different inhibitory activity. The first one showed a weak affinity for PTP1B but behaved as a potent α-glucosidase inhibitor. The latter showed a reduced affinity for both PTP1B α-glucosidase when compared to (**229**). Finally, Clausenaline F (**223**), bearing two methoxy groups on C1 and C2, and Clausines B (**232**), possessing 2 methoxy groups on C6 and C8, showed a decreased affinity for PTP1B in comparison with (**229**). However, Clausines B resulted a better α-glucosidase inhibitor than Clausenaline F. The replacement of the carboxyl group of (**229**) with a formyl group leads to 3-Formyl-1-hydroxycarbazole (**220**), which showed an affinity for PTP1B similar to (**229**) but an increased affinity for α-glucosidase. The loss of the OH on C1 of (**231**) leads to Clausenanisine D (**227**), which showed a higher IC_50_ value for PTP1B but a similar affinity for the α-glucosidase compared to (**220**). Changing the position of one or both OH groups present on (**220**) we obtain Clauszoline N (**234**) and Clauszoline M (**233**). The affinity of these compounds is similar, but that for α-glucosidase differs, (**233**) being a very bad inhibitor of this enzyme. The addition of one methoxy group on C6 of 3-Formyl-1-hydroxycarbazole (**220**) leads to Clausine I (**230**), which possessed a similar affinity for PTP1B as (**220**) but an enhanced affinity for α-glucosidase compared to 3-Formyl-1-hydroxycarbazole. Finally, the introduction of two methoxy groups on C6 and C8 of Clauszoline M (**233**) leads to Clausines B (**232**), a molecule that showed an affinity for PTP1B similar to that of (**233**) but a high affinity for α-glucosidase.

### 2.12. Others

Several different molecules were obtained from different natural sources, such as *Euonymus alatus* (Thunb.) Sieb. [38], *Hizikia fusiformis* [43], *Pueraria lobata* [49], *Paeonia lactiflora* [53], *Ecklonia stolonifera* and *Eisenia bicyclis* [57], and *Artemisia capillaris* [67] (Table 12).

Some of these, such as Phloroglucinol (**243**), Dioxinodehydroeckol (**240**), Eckol (**241**), and Phlorofucofuroeckol-A (**242**), are naturally occurring phloroglucinol polymers. Phlorofucofuroeckol-A exhibited the higher inhibitory activity, with IC_50_ values for PTP1B and α-glucosidase of 0.56 and 1.37 µM, respectively. Moreover, kinetic analyses revealed that Phlorofucofuroeckol-A acts as a non-competitive inhibitor of both PTP1B and α-glucosidase. Interestingly, the affinity of Eckol, Dioxinodehydroeckol, and Phloroglucinol is progressively decreasing, suggesting that the phloroglucinol polymers ensure a stronger interaction with both targets, probably due to the high number of hydrogen bonds that are generated between the hydroxyl groups of molecules and the amino acids of the target.

*p*-propoxybenzoic acid (**244**) showed a slight decrease of affinity for both targets (IC_50_ values for PTP1B and α-glucosidase were 14.8 and 10.5 µM, respectively), while Sargachromenol (**245**) and Sargaquinoic acid (**246**), showed a similar affinity for PTP1B but a decreased affinity for α-glucosidase in respect to (**244**). All other compounds showed a weak affinity for both targets.

## 3. From Natural World to Bench-Side

What emerges from the data reported in this review is that the natural world can provide a large number of chemically different scaffold molecules active on both PTP1B and α-glucosidase. The list includes a large number of good inhibitors—molecules showing IC_50_ values in the low micromolar range (0.1–20 µM) for both targets—and other compounds with IC_50_ values that gradually increase up to the value of 440 µM for the PTP1B and 600 µM for the α-glycosidase (Figure 5 and Table 13).

It is interesting to note that the 250 known MDLs show a greater affinity for PTP1B than for α-glucosidase, as evidenced by the fact that the average IC_50_ values for the enzymes are 6.0 ± 4.1 µM and 42.9 ± 52.0 µM, respectively. Although this difference could seem an obstacle to the development of balanced dual PTP1B/α-glucosidase inhibitors, it may not represent a true problem, especially considering the different localization and physiological role of the two enzymes. The α-glucosidase localizes in the intestinal lumen while PTP1B in the cytosol of human cells [69]. This first consideration suggests that α-glucosidase targeting does not require very potent inhibitors, and significant inhibition of this enzyme can easily be achieved by taking oral drug during or immediately after a meal. By ensuring protection from the acidic pH of the stomach, MDLs could reach the intestine intact and target the enzyme. Furthermore, the lower potency of MDLs for α-glucosidase could be easily compensated by increasing drug dosage. As a precautionary measure, it could be of great importance to evaluate the ability of these molecules to inhibit the activity of intestinal glucose transporters in order to avoid the risk of inducing a hypoglycemic condition [70]. Conversely, targeting PTP1B is a more complex matter. In fact, MDLs designed to act on PTP1B should have good bioavailability to ensure their absorption through the intestinal wall and their rapid uptake by cells that compose liver, muscle, adipose tissue, pancreas, and brain, where this enzyme acts as a negative regulator of the insulin receptor. Moreover, considering that mammalian cells express different types of tyrosine phosphatases, each of which carries out important regulatory functions, a good/high specificity of MDLs for PTP1B is essential to avoid the induction of severe side effects. Unfortunately, in most cases, information about the specificity of such molecules are not yet available. Conversely, relatively more informations are available about the action mechanism of natural MDLs (Table 14). It is interesting to note that several of the most effective MDLs identified act as non-competitive inhibitors of PTP1B, suggesting that they interact with sites different from the active site of enzyme. Non-competitive and allosteric PTP1B inhibitors actually represent the most promising molecules for developing new antidiabetic drugs since they could ensure a more adequate and specific action, thereby reducing the risk of inducing negative side effects [71].

From a careful analysis of the data, it is possible to obtain other important information regarding the properties of the molecules with dual inhibitory activity. The “α-glucosidase IC_50_/PTP1B IC_50_” ratio allows us to evaluate the relative potency of MDLs toward both targets. Interestingly among all compounds, 110 of them show an IC_50_ ratio comprised between 0.5 and 1, while the other 35 compounds show a ratio value comprised between 1 and 2 (Table 15). This finding demonstrates that about 59% of identified molecules can be considered naturally balanced inhibitors even if each of these molecules differ from the others in inhibitory power. Nevertheless, we also found compounds showing a different balancing ratio, including molecules with a greater affinity for PTP1B in respect to α-glucosidase and vice versa. For this reason, we think that such data could be precious for researchers working to produce new PTP1B/α-glucosidase inhibitors with different kinetic properties and structure to be used as scaffold molecules for the design of new MDLs with antidiabetic activity. Otherwise, some of these molecules could be used to generate semisynthetic derivatives characterized by greater bioavailability or specificity for selected targets [6].

Considering that the chemical properties can influence both the efficacy and metabolic fate of each molecule, the choice of the most appropriate scaffolds remains one of the most important steps of the entire design and synthesis process. A seminal study conducted by Lipinski and coworkers lead to formulation of general criteria useful to predict in-vivo bioavailability and degree of oral absorption of a molecule. The molecules that are most likely to be absorbed are characterized by a molecular weight lower than 500 Da, a miLogP value comprised in the 0–5 range, a number of hydrogen bond acceptors <10, a number of hydrogen bond donors <5, a TPSA (topological polar surface area) value <140 Å, and a number of rotatable bonds <10–20 [72]. Based on this information, we decided to analyze the properties of the 60 most potent compounds among those reviewed to predict their bioavailability. Interestingly, we found that, on average, the compounds meet the criteria mentioned before (Table 16). This result reinforces the hypothesis that most of these compounds possess a good bioavailability and can be considered promising lead molecules for the development of new MDLs.

Besides the influence on bioavailability, we wondered if the aforementioned chemical-physical parameters could influence the activity of MDLs toward designed targets. To give an answer, we analyzed the correlation between the IC_50_ values for both enzymes and each parameter described in Table 16 (Figure 6 and Figure 7).

Interestingly, taking into account the PTP1B inhibitory activity, we found that the TPSA, the number of hydrogen donors and acceptors, the complexity, and the molecular weight of molecules are inversely related to IC_50_ value, while the miLogP shows a direct correlation with the calculated IC_50_ value. A weak correlation was observed between the number of rotatable bonds and the IC_50_ value. This evidence suggests that the extension of the molecules and their ability to form hydrogen bonds with the atoms of the enzyme play an important role in determining their interaction strength with PTP1B, while the hydrophobicity of the molecules represents an unfavorable factor. However, different results were obtained analyzing the IC_50_ values for α-glucosidase. Parameters such as TPSA, number of hydrogen donors and acceptors, complexity, and molecular weight have a poor correlation with the IC_50_ value, while the parameter miLogP seems inversely correlated with the inhibitory power of the molecules. Finally, the number of rotatable bonds appears directly related to the IC_50_ value for α-glucosidase. These data suggest that this enzyme preferentially interacts with rigid molecules characterized by a hydrophilic profile even if the interaction of MDLs with the enzyme is not strongly influenced by formation of hydrogen bond or by molecular weight. This is in accordance with the fact that this enzyme works in an aqueous extracellular medium and binds glucose polymers, molecules showing a relatively high rigidity.

## 4. Mechanism of Action and In-Vivo Activity of Some Natural PTP1B/α-Glucosidase Inhibitors

Most of the studies carried out to identify new double inhibitors of PTP1B/α-glucosidase have been focused on the definition of the chemical structure and on the determination of the IC_50_ values of the molecules. Rarely, researchers have also been able to determine also the mechanism of action and in-vivo effectiveness of the isolated compounds. Below, we have summarized the data obtained from the literature concerning some of the most potent double inhibitors of PTP1B/α-glucosidase characterized so far.

Phlorofucofuroeckol-A (**242**), Dieckol (**139**), and 7-Phloroeckol (**122**) are phlorotannins extracted by the edible brown algae arame (*Ecklonia bicyclis*) and turuarame (*Ecklonia stolonifera*).



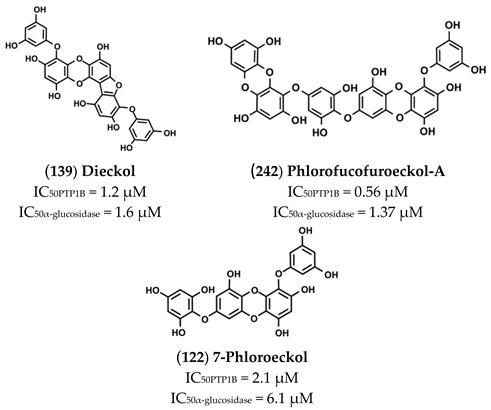



Kinetic analyses revealed that these compounds acted as non-competitive inhibitors of PTP1B, while only Phlorofucofuroeckol-A and 7-Phloroeckol behaved as non-competitive inhibitors of α-glucosidase. Conversely, Dieckol acted as competitive inhibitor of the latter. A recent investigation on rat insulinoma cells showed that treatment with Dieckol reduced oxidative stress and apoptosis caused by exposition of cells to high glucose levels, suggesting that this compound could protect β-pancreatic cells against damages induced by hyperglycemia [73].

A recent in-vivo study demonstrated that Phlorofucofuroeckol-A delayed intestinal absorption of dietary carbohydrates in diabetic mice models, suggesting that this molecule could be used to generate new drugs able to reduce post-prandial glucose levels in diabetic patients [74]. High blood glucose levels increase the production of advanced glycation end products that, in turn, contribute to the onset of numerous diabetes-related complications, such as nephropathy, retinopathy, atherosclerosis, and neurodegenerative diseases. Recently, Su Hui Seong et al. demonstrated that Phlorofucofuroeckol-A inhibits non-enzymatic insulin glycation and Aβ aggregation. This finding suggests that this molecule could be used to preserve insulin function and prevent aggregation of the Aβ peptide, thus maintaining the vitality of neurons and avoiding Aβ-mediated brain damage [75]. Besides, it has been demonstrated that Phlorofucofuroeckol-A acts as a potent non-competitive inhibitor of human monoamine oxidase-A, confirming that this compound could be useful to prevent neuronal disorders [76]. A further study showed that Phlorofucofuroeckol-A is effective in contrasting negative effects induced by high fatty diet as well as in reducing leptin resistance in hypothalamic neurons and microglia. This evidence suggests that such compound could be used to fight leptin resistance and to control weight in obese subjects [77].

Results of studies conducted on diabetic mice models confirmed the antidiabetic activity of Dieckol (**139**) in vivo. Intra peritoneal injections (10–20 mg/Kg for 14 days) of Dieckol in C57BL/KsJ-db/db diabetic mice lead to a significant reduction of both blood glucose and serum insulin levels as well as of body weight. Moreover, it has been reported that treated mice showed either increased phosphorylation levels of AMPK (5′ adenosine monophosphate-activated protein kinase) or Akt (Akt kinase) and a concomitant increase of the activity of several antioxidant enzymes [78]. Seung-Hong Lee and co-workers showed that the treatment of type 2 diabetic mice with a Dieckol-rich extract from *Ecklonia cava* caused a significant reduction of blood glycosylated hemoglobin and plasma insulin levels and an improvement of glucose tolerance. Furthermore, diabetic mice treated with Dieckol-rich extract showed a reduction of both plasma and hepatic lipids levels compared to control mice group [79]. Similar results were obtained from tests carried out on eighty pre-diabetic male and female adults. In this case, the administration of Dieckol-rich extract (1500 mg per day for 12 weeks) caused a significant reduction of postprandial glucose, insulin, and C-peptide levels compared to control group. Moreover, no significant adverse events or changes of biochemical and hematological parameters were observed during treatment. Taken together, these results confirmed that algae extracts rich in Dieckol and other phlorotannins possess antidiabetic activity, and their consumption may contribute to regularize glycemia in prediabetic and diabetic subjects [80].

Mulberrofuran G (**170**), Albanol B (**122**), and Kuwanon G (**209**) are flavonoids extracted from root bark of *Morus Alba* Linn.



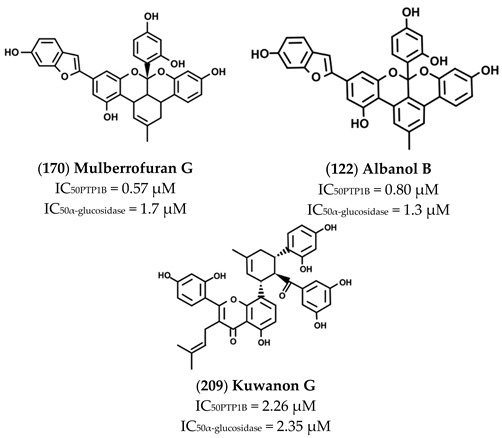



Paudel P. et al. demonstrated that such compounds acted as potent mixed-type enzyme inhibitors against PTP1B and α-glucosidase [59]. In accordance with the results of kinetic analyses, docking in silico revealed that these compounds interacted with both the catalytic and an allosteric site present on PTP1B surface. Similar results were obtained analyzing the interaction mode of compounds with α-glucosidase enzyme, thereby confirming the mixed-type inhibition mode. Studying the effects on liver cells, the authors reported that treatment with these compounds decreased the expression of PTP1B and stimulated glucose uptake, suggesting that they act as insulin-sensitizing agents [59]. Several phytochemicals extracted from leaves and fruits of different members belonging to the Moraceae family showed anti-obesity and anti-diabetic activity, enhancing insulin signaling pathway, glucose uptake, and inhibiting hepatic gluconeogenesis [81,82].

Like the above compounds, also Morusalbin D (**169**), Albasin B (**127**), and Macrourin G (**210**) were extracted from *Morus Alba* Linn.



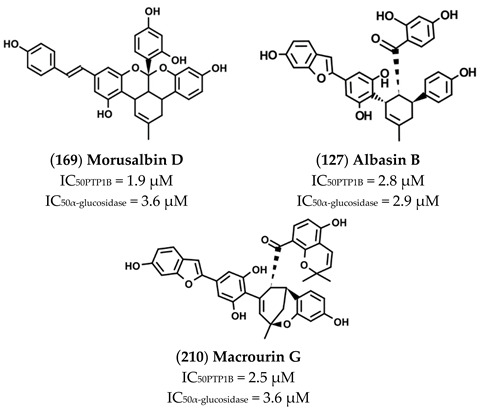



Manh Tuan Ha and co-workers demonstrated that these compounds act as non-competitive inhibitors of PTP1B and as competitive inhibitors of α-glucosidase [56]. Besides, in-silico docking analyses revealed that all three compounds are docked into the allosteric binding site previously described by Wiesmann et al. in 2004 [65]. Although no data have been produced to confirm the in-vivo activity of such compounds, such results could inspire the synthesis of new MDLs for treatment of type 2 diabetes [56].

Ugonin J (**190**) together with other similar derivatives are flavonoids with cyclohexyl motif extracted from the rhizome part of *Helminthostachys zeylanicawas*. Abdul Bari Shah and co-workers demonstrated that this compound acts as a competitive inhibitor of PTP1B and non-competitive inhibitor of α-glucosidase [60].



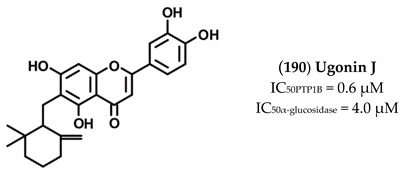



Interestingly, deeper analyses revealed that Ugoning J behaves as a reversible slow-binding inhibitor on PTP1B. In-silico docking analyses confirmed that Ugonin J interacted with residues located in the active site of PTP1B, thereby confirming its nature of competitive inhibitor [60]. An in-vivo study conducted on C57BL/6 J mice fed a high-fat diet showed that Ugonin J treatment promoted lipid clearance increasing phosphorylation levels of both AMPK, ACC (acetyl-CoA carboxylase), and upregulating th expression of CPT-1 (carnitine palmitoyltransferase-1). Moreover, it has been observed that treatment of liver cells with Ugonin J increased Akt activity and increased insulin secretion from β cells during an acute insulin-secretion tests [83].

The xanthones α-Mangostin (**50**), γ-Mangostin (**49**), and Cratoxanthone A (**56**) were extracted from *Cratoxylum cochinchinense* Lour. Results of kinetic analyses revealed that these compounds behave as mixed-type inhibitors of α-glucosidase but as competitive inhibitors of PTP1B [56].



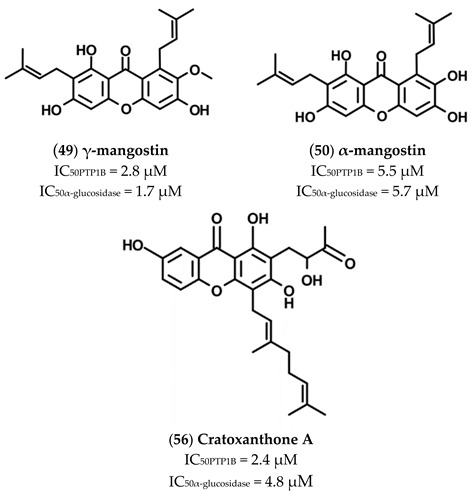



Alaternin (**100**) and Emodin (**110**) are two anthraquinones extracted from *Cassia obtusifolia*.



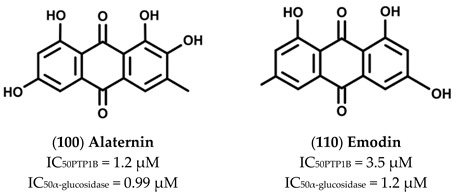



Alaternin acted as a PTP1B-competitive inhibitor, and its interaction with the active site of enzyme was confirmed also by in-silico docking analyses. Besides, it has been demonstrated that Alaternin behaved as a mixed-type inhibitor of α-glucosidase. Finally, the authors of the study showed that both compounds are well tolerated by HepG2 cells and enhanced insulin-stimulated glucose uptake [45].

## 5. Conclusions

MDLs are considered promising alternatives to traditional drugs for the treatment of multifactorial disease. However, the production of new MDLs is not a simple matter. The main obstacle that researchers face in the initial phase of design and synthesis of new drugs is the selection of appropriate scaffold molecules that exhibit at least one initial activity against the selected targets. Sometimes this phase can be very expensive and time consuming, two factors that can lead to the failure of the entire project. In this contest, there is a large consensus in considering the natural world an important source of scaffold molecules to be used for design new MDLs. Data reported in this review demonstrated that many natural molecules possess intrinsic dual α-glucosidase/PTP1B-inhibitory activity. Although most of these possess sub-optimal properties, such as low bioavailability and unfavorable pharmacokinetic or a lowspecificity, semi- or fully synthetic superior derivatives could be easily obtained starting from them. This approach could allow researchers to quickly overcome the first experimental phase, allowing them to focus on the optimization phase aimed at balancing the activity of the molecules toward the biological targets, increasing theirspecificityor bioavailability.

The data we reported confirmed that many natural molecules, including some lignanamides, xanthones, anthraquinones, and several phenolic compounds, show high/balanced inhibitory activity toward PTP1B and α-glucosidase, making these as interesting lead compounds for synthesis of new MDLs. Most of these compounds are characterized by the presence of several hydroxylated phenolic rings or aliphatic chains. What often emerges from SAR analyses is that the number and the position of hydroxyl groups or aliphatic chains are essential to ensure a close interaction between the inhibitors and both enzymes and that by changing one of these parameters, it is possible to influence both inhibitory power and specificity for targets. Moreover, we found that complexity and molecular weight are two parameters strictly related with the IC_50_ values of molecules, confirming that the higher the number of OH groups, the higher the affinity for targets. However, small molecules such as Alaternin, Ugonin J, or α-Mangostin also showed high affinity for both targets, suggesting that it is possible to reach a high inhibitory activity also starting from small scaffold molecules. This is another interesting aspect concerning the mechanism of action of these molecules. Data reported in Table 14 shows that smaller molecules mainly behave as competitive inhibitors for PTP1B and bigger ones as non-competitive inhibitors. This evidence suggests that small molecules bearing a hydroxylated phenyl ring could mimic the phenolic structure of the natural substrate, the phosphotyrosine, and for this reason, they could interact more easily with the active site of the enzyme. Conversely, larger molecules, due to their steric hindrance, would not be able to access the active site but could bind to allosteric sites on the surface of the enzyme.

To the best of our knowledge, few synthetic PTP1B/α-glucosidase inhibitors have been produced still to date, and none of these has been evaluated in clinical trials to date, which is probably due to the fact that development of such kinds of molecules is still in its infancy. We hope that the information reported in this review will be useful to researchers dedicated to the design and synthesis of novel dual PTP1B/α-glucosidase inhibitors to be used routinely for the treatment of patients affected by T2D, obesity, and metabolic syndrome.

## Figures and Tables

**Figure 1 molecules-26-04818-f001:**
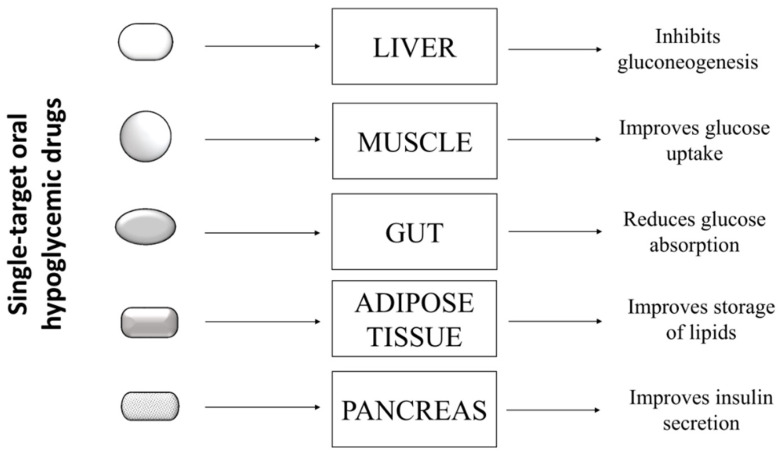
Antidiabetic strategy based on the mono-drug therapy: each antidiabetic drug is administered alone and works on a specific target.

**Figure 2 molecules-26-04818-f002:**
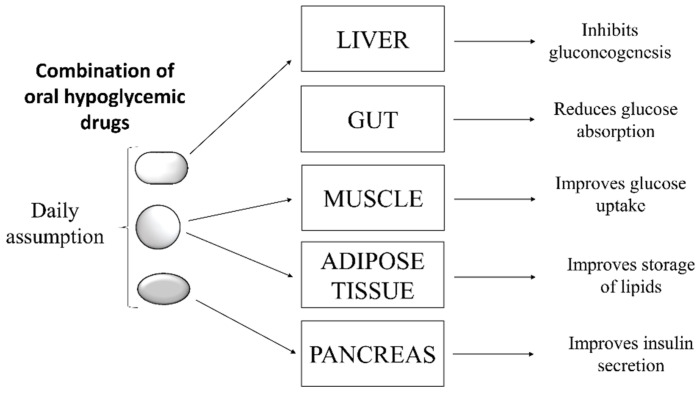
Antidiabetic strategy based on combination therapy: patients are treated with two or more antihyperglycemic oral drugs also administered at different times of the day.

**Figure 3 molecules-26-04818-f003:**
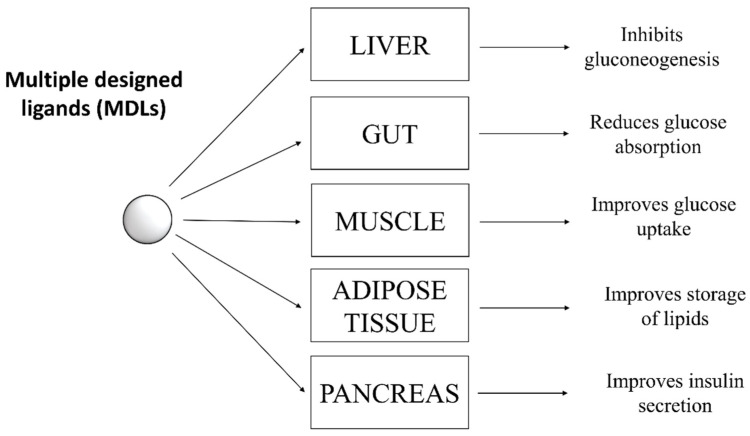
Antidiabetic strategy based on administration of multiple designed ligands: a single drug regulates different specific targets.

**Figure 4 molecules-26-04818-f004:**
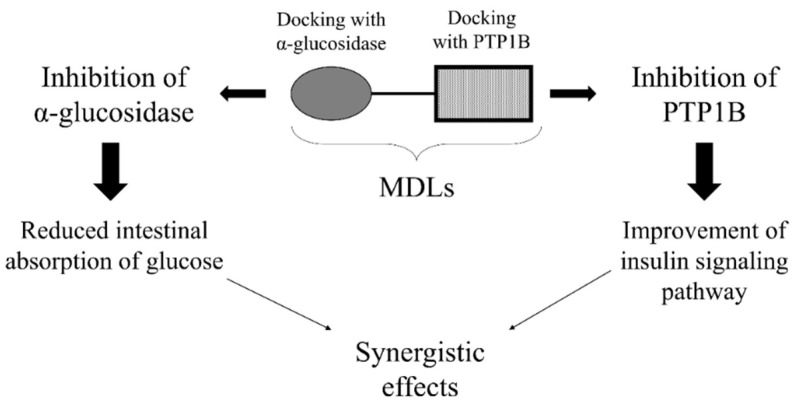
Mechanism of action of a dual PTP1B/α-glucosidase inhibitor. Such molecules possess a structure able to interact with both targets, thereby leading to their inhibition.

**Figure 5 molecules-26-04818-f005:**
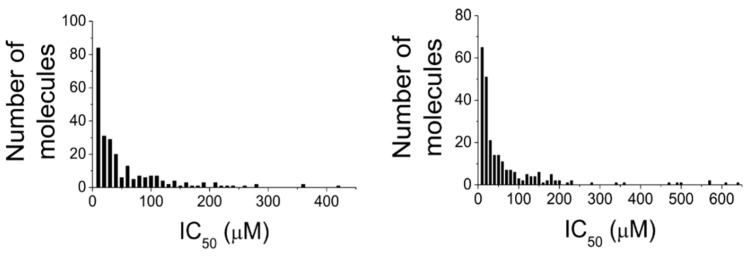
Distribution of the IC_50_ values calculated for both PTP1B (**left**) and α-glucosidase (**right**). The IC_50_ values are categorized as shown in Table 13.

**Figure 6 molecules-26-04818-f006:**
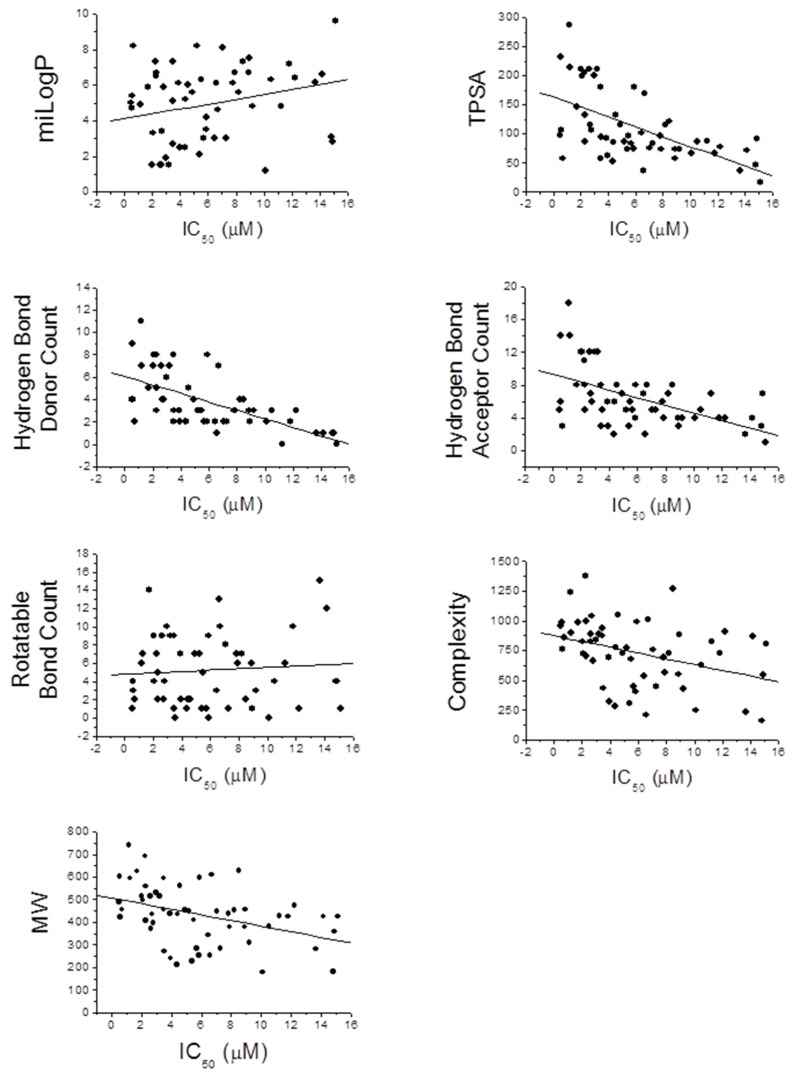
Relationship between the chemical-physical properties and the IC_50_ values for PTP1B of the best sixty compounds. Chemical-physical data were recovered from PubChem database and analyzed using OriginPro 2021 Software.

**Figure 7 molecules-26-04818-f007:**
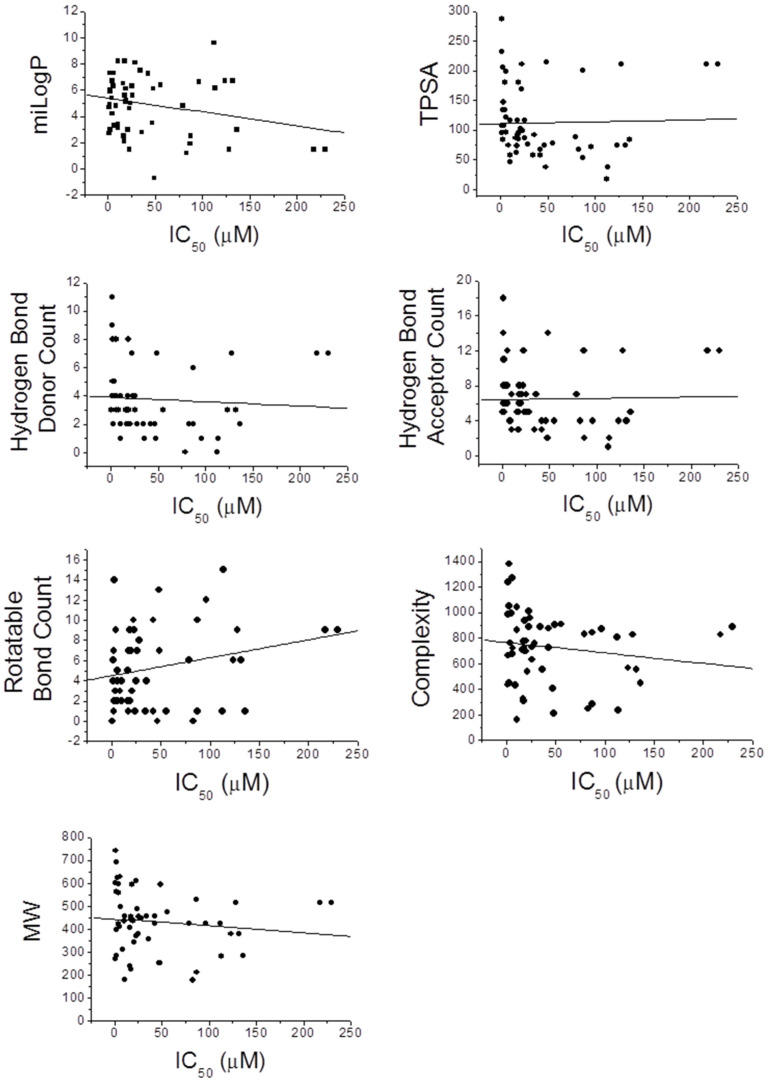
Relationship between the chemical-physical properties and the IC_50_ values for α-glucosidase of the best sixty compounds. Chemical-physical data were recovered from PubChem database and analyzed using OriginPro 2021 Software.

**Table 1 molecules-26-04818-t001:** Coumarins.

Compound		IC_50_ (µM)		Reference
		PTP1B	α-Glucosidase	
**1**	(+)-*trans*-decursidinol	2.33 ± 0.07	11.32 ± 0.56	[36]
**2**	Pd-C-I	4.32 ± 0.12	17.40 ± 0.33	[36]
**3**	Pd-C-II	6.17 ± 0.31	24.74 ± 0.89	[36]
**4**	2′-Isopropyl psoralene	10.78 ± 0.17	85.82 ± 0.99	[37]
**5**	4′-Hydroxy Pd-C-III	5.39 ± 0.19	77.30 ± 1.07	[37]
**6**	Pd-C-III	11.98 ± 0.43	36.77 ± 1.04	[36]
**7**	4′-Methoxy Pd-C-I	6.62 ± 0.77	89.19 ± 0.77	[37]
**8**	Euonymalatus	13.7 ± 0.2	9.1 ± 0.5	[38]
**9**	Esculetin	10.1 ± 1.9	82.92 ± 4.81	[39]
**10**	6-Methoxy artemicapin C	27.61 ± 8.30	563.75 ± 6.55	[39]
**11**	Daphnetin	164.50 ± 7.44	560.20 ± 19.60	[39]
**12**	Decursidin	11.22 ± 0.39	79.09 ± 0.11	[37]
**13**	Decursinol	58.90 ± 1.07	65.29 ± 0.81	[37]
**14**	Scopoletin	227.28 ± 26.71	159.16 ± 11.71	[39]
**15**	Selaginolide A	7.40 ± 0.28	7.52 ± 0.37	[40]
**16**	Umbelliferone	274.86 ± 20.27	633.94 ± 23.78	[39]
**17**	Umbelliferone 6-carboxylic	7.98 ± 0.91	172.10 ± 0.19	[37]

**Table 2 molecules-26-04818-t002:** Lignans.

Compound		IC_50_ (µM)		Reference
		PTP1B	α-Glucosidase	
**18**	(−)-Olivil 4′[4-*O*-*β*-d-glucoside	27.5 ± 0.6	40.3 ± 1.6	[41]
**19**	(−)-Olivil 4-*O*-*β*-d-glucoside	51.0 ± 1.1	50.1 ± 2.1	[41]
**20**	(+)-8′-Hydroxypinoresinol 4-*O*-*β*-d-glucoside	28.7 ± 1.0	56.6 ± 1.7	[41]
**21**	(+)-8-Hydroxypinoresinol 4-*O*-β-d-glucoside	31.8 ± 1.0	57.2 ± 1.6	[41]
**22**	(+)-Pinoresinol 4-*O*-*β*-d-glucoside	129.5 ± 2.1	61.7 ± 2.4	[41]
**23**	(+)-Syringaresinol 4-*O*-*β*-d-glucoside	120.4 ± 2.5	69.5 ± 2.3	[41]
**24**	(7*S*,8*R*,7′*S*,8′*S*)-4,9,4′,7′-Tetrahydroxy-3,3′-dimethoxy-7,9′-epoxylignan-4-*O*-*β-*d-glucoside	51.8 ± 1.1	42.5 ± 2.1	[41]
**25**	3,3′-Demethyl-heliotropamide	11.27 ± 1.00	14.48 ± 0.55	[42]
**26**	Cannabisin A	3.47 ± 0.49	18.67 ± 0.88	[42]
**27**	Cannabisin B	5.89 ± 0.83	4.56 ± 0.21	[42]
**28**	Cannabisin C	6.69 ± 0.87	22.56 ± 2.37	[42]
**29**	Cannabisin D	29.17 ± 2.76	187.08 ± 7.42	[42]
**30**	Cannabisin F	1.71 ± 0.28	2.99 ± 0.15	[42]
**31**	Cannabisin I	2.01 ± 0.13	1.50 ± 0.07	[42]
**32**	Cedrusin	174.19 ± 5.44	133.84 ± 3.86	[43]
**33**	Limoniumin A	6.66 ± 0.43	2.93 ± 0.16	[42]
**34**	Limoniumin B	5.05 ± 2.44	2.37 ± 0.23	[42]
**35**	Limoniumin C	4.65 ± 1.34	20.55 ± 0.30	[42]
**36**	Limoniumin D	9.77 ± 0.74	10.22 ± 0.25	[42]
**37**	Limoniumin E	5.36 ± 0.96	29.16 ± 2.61	[42]
**38**	Limoniumin F	8.12 ± 0.88	25.23 ± 1.17	[42]
**39**	Limoniumin H	3.24 ± 0.23	70.11 ± 4.17	[42]
**40**	Limoniumin I	7.65 ± 2.15	9.30 ± 1.07	[42]
**41**	Viburmacroside A	91.5 ± 1.9	16.7 ± 1.0	[41]
**42**	Viburmacroside B	92.9 ± 1.8	17.1 ± 1.3	[41]
**43**	Viburmacroside C	25.8 ± 0.9	16.1 ± 1.0	[41]
**44**	Viburmacroside D	8.9 ± 0.5	9.9 ± 0.6	[41]
**45**	Viburmacroside E	102.3 ± 2.2	68.6 ± 2.5	[41]
**46**	Viburmacroside F	101.4 ± 2.1	66.8 ± 2.3	[41]
**47**	Viburmacroside G	44.7 ± 1.3	56.9 ± 1.9	[41]
**48**	Viburmacroside H	43.6 ± 1.2	64.4 ± 2.0	[41]

**Table 3 molecules-26-04818-t003:** Xanthones.

Compound		IC_50_ (µM)		Reference
		PTP1B	α-Glucosidase	
**49**	γ-Mangostin	2.8	1.7	[44]
**50**	α-Mangostin	5.5	5.7	[44]
**51**	1,3,7-Trihydroxy-2,4-diisoprenylxanthone	10.5	25.5	[44]
**52**	7-Geranyloxy-1,3-dihydroxyxanthone	35.1	20.6	[44]
**53**	Caratoxanthone A	2.4	4.8	[44]
**54**	Cochinchinone Q	52.5	72.7	[44]
**55**	Cochinechinone A	5.2	17.4	[44]
**56**	Cochinxanthone A	25.2	13.7	[44]
**57**	Cratoxanthone E	7.2	16.1	[44]
**58**	Cratoxanthone F	12.5	10.1	[44]
**59**	Cratoxylone	24.1	30.7	[44]
**60**	Pruniflorone S	7.05	28.4	[44]
**61**	Toralactone gentiobioside	81.15 ± 0.15	37.60 ± 0.79	[45]

**Table 4 molecules-26-04818-t004:** Terpenes.

Compound		IC_50_ (µM)		Reference
		PTP1B	α-Glucosidase	
**62**	3-*O*-[*β*-d-Glucopyranosyl(1→3)-*β*-d-gluco pyranosyl]-caulophyllogenin 28-*O*-*β*-d-glucopyranosyl ester	20.37 ± 0.46	2.96 ± 0.32	[46]
**63**	3-*O*-*α*-l-Arabinopyranosyl echinocystic acid	11.22 ± 0.33	0.73 ± 0.39	[46]
**64**	18α-Glycyrrhetinic acid	10.40 ± 0.75	113.30 ± 0.70	[43]
**65**	18β-Glycyrrhetinic acid	26.07 ± 0.59	128.72 ± 3.88	[43]
**66**	20(*R*)-25-Methoxydammarane-3*β*,12*β*, 20-tetrol	16.54	3.15 ± 0.07	[47]
**67**	20(*R*)-Dammarane-*3β*,6*α*,12*β* 20, 25-pentol	10.07	2.5 ± 0.15	[47]
**68**	20(*R*)-Protopanaxadiol	21.27	35.26 ± 0.15	[47]
**69**	20(*R*)-Protopanaxatriol	17.98	3.57 ± 0.22	[47]
**70**	20(*S*)-Panaxatriol	21.02	9.15 ± 0.72	[47]
**71**	24*R*-Methyllophenol	15.1 ± 0.7	183.1 ± 1.2	[38]
**72**	Betulinic acid	16.05 ± 1.34	>100	[48]
**73**	Corosolic acid	12.21 ± 0.46	55.54 ± 3.14	[48]
**74**	Epi-lupeol	28.4 ± 1.2	133.1 ± 0.9	[38]
**75**	Jacoumaric acid	11.93 ± 0.31	>100	[48]
**76**	Lupenone	15.11 ± 1.23	112.36 ± 0.18	[49]
**77**	Lupeol	38.89 ± 0.17	176.35 ± 0.21	[49]
**78**	Maslinic acid	25.73 ± 0.82	>100	[48]
**79**	Oleanolic acid	8.92 ± 0.46	34.29 ± 2.58	[48]
**80**	Tormentic acid	0.50 ± 0.06	23.8 ± 0.4	[50]
**81**	Ursolic acid	3.47 ± 0.02	42.4 ± 0.7	[50]

**Table 5 molecules-26-04818-t005:** Glycosides.

Compound		IC_50_ (µM)		Reference
		PTP1B	α-Glucosidase	
**82**	1-Feruloyl-*β*-d-glucoside	18.4 ± 0.3	169.7 ± 1.5	[38]
**83**	Daidzin	207.68 ± 0.45	485.73 ± 2.81	[49]
**84**	Didymin	1.23 ± 0.11	48.77 ± 1.02	[51]
**85**	Kaempferol-3-*O*-*α*-l-rhamnoside	12.16 ± 0.02	28.5 ± 0.1	[50]
**86**	Apigenin-7-*O*-*β*-d-glucuronide-6″-methyl ester	14.35 ± 0.76	103.3 ± 1.1	[50]
**87**	Quercetin-3-*O*-*β*-d-glycoside	27.73 ± 1.54	29.6 ± 0.9	[50]
**88**	Puerarin	115.81 ± 1.72	147.30 ± 1.33	[49]
**89**	Puerarol B-2-*O*-glucoside	258.15 ± 0.33	609.43 ± 1.78	[49]
**90**	Quercetin-3-(2-*E*-sinapoyl)-*O*-glucopyranoside	64.92 ± 5.80	4.96 ± 0.61	[52]
**91**	Quercetin-3-*O*-(6′′-*O*-E-*p*-feruloyl)-*β*-d-glucopyranoside	39.40 ± 4.55	10.86 ± 0.91	[52]

**Table 6 molecules-26-04818-t006:** Fatty Acids.

Compound		IC_50_ (µM)		Reference
		PTP1B	α-Glucosidase	
**92**	(*Z*)-Hexadec-12-enoic acid	6.59 ± 0.09	48.05 ± 3.37	[43]
**93**	(*Z*)-Octaec-9-enoic acid	13.65 ± 0.49	113.44 ± 2.47	[43]
**94**	Methyl 2-hydroxyl tricosanoate	36.39 ± 1.72	112.8 ± 1.7	[50]
**95**	Palmitic acid	0.10 ± 0.03	45.5 ± 1.5	[50]
**96**	(7*Z*,10*Z*,13*Z*)-Octadeca-7,10,13-trienoic acid	13.58 ± 0.10	111.51 ± 1.44	[43]
**97**	(7*Z*,9*Z*,11*Z*,13*Z*)-Eicosa-7,9,11,13-tetraenoic acid	10.68 ± 0.17	34.85 ± 2.39	[43]
**98**	(8*Z*,11*Z*,14*Z*)-Heptadeca-8,11,14-trienoic acid	11.51 ± 0.52	43.90 ± 0.77	[43]

**Table 7 molecules-26-04818-t007:** Anthraquinones.

Compound		IC_50_ (µM)		Reference
		PTP1B	α-Glucosidase	
**99**	2-Hydroxyemodin-1 methylether	5.22 ± 0.29	5.65 ± 0.20	[45]
**100**	Alaternin	1.22 ± 0.03	0.99 ± 0.02	[45]
**101**	Aloe-emodin	56.01 ± 0.76	1.40 ± 0.27	[45]
**102**	Aurantio-obtusin	27.19 ± 0.31	41.20 ± 0.17	[45]
**103**	Cassiaside	48.55 ± 1.27	129.23 ± 0.98	[45]
**104**	Cassitoroside	103.89 ± 1.22	172.59 ± 0.74	[45]
**105**	Chryso-obtusin	14.88 ± 0.77	36.01 ± 0.89	[45]
**106**	Chryso-obtusin-2-glucoside	39.34 ± 1.07	178.85 ± 0.55	[45]
**107**	Chrysophanol	5.86 ± 0.99	46.81 ± 0.12	[45]
**108**	Chrysophanol tetraglucoside	103.89 ± 1.22	228.79 ± 0.91	[45]
**109**	Chrysophanol triglucoside	80.17 ± 1.77	197.06 ± 1.09	[45]
**110**	Emodin	3.51 ± 0.15	1.02 ± 0.01	[45]
**111**	Gluco-aurantio obtusin	31.30 ± 0.97	142.19 ± 1.22	[45]
**112**	Gluco-obtusifolin	53.35 ± 0.44	23.77 ± 0.72	[45]
**113**	Obtusifolin	35.27 ± 0.98	142.12 ± 0.77	[45]
**114**	Obtusin	6.44 ± 0.22	20.92 ± 0.41	[45]
**115**	Physcion	7.28 ± 0.49	2.38 ± 0.77	[45]
**116**	Questin	5.69 ± 0.47	136.19 ± 0.01	[45]

**Table 8 molecules-26-04818-t008:** Phenolic compounds.

Compound		IC_50_ (µM)		Reference
		PTP1B	α-Glucosidase	
**117**	(−)-7*α*,8*α*-*cis*-ε-viniferin	97.74 ± 1.68	59.16 ± 2.91	[53]
**118**	(3*S*,5*S*,6*S*,7*S*)-6-hydroxycalyxin F	100.3 ± 8.6	10.1 ± 0.6	[54]
**119**	4′-(6,6-dimethyl-5-hydroxyl-2-methylenecyclohexylmethyl)-3′,5′,6-trihydroxy-2-arylbenzofuran	38.1 ± 1.8	11.9 ± 1.3	[55]
**120**	5′-(1″,1″-dimethylallyl)-5,7,2″,4″-tetrahydroxyflavone	56.18 ± 4.46	11.04 ± 0.76	[56]
**121**	6,7-dimethoxy-2′,4′-dihydroxyisoflavone	11.08 ± 0.92	55.73 ± 2.58	[40]
**122**	7-Phloroeckol	2.09 ± 0.09	6.13 ± 0.30	[57]
**123**	Akebonic acid	57.8	73.5	[58]
**124**	Albafuran A	7.9 ± 0.4	123.6 ± 7.1	[55]
**125**	Albafuran B	8.9 ± 1.1	131.9 ± 4.5	[55]
**126**	Albanol B	0.80 ± 0.02	1.31 ± 0.01	[59]
**127**	Albasin B	2.80 ± 0.19	2.90 ± 0.10	[56]
**128**	Alpinnanin A	35.9 ± 0.4	11.4 ± 1.8	[54]
**129**	Alpinnanin B	32.6 ± 1.6	19.4 ± 1.6	[54]
**130**	Calycosin	273.23 ± 1.85	6.84 ± 1.58	[49]
**131**	Calyxin B	22.5 ± 3.1	13.1 ± 0.3	[54]
**132**	Calyxin H	186.8 ± 26.1	8.9 ± 1.7	[54]
**133**	Calyxin L	96.7 ± 3.5	4.3 ± 0.2	[54]
**134**	Carasiphenol A	137.43 ± 13.48	32.88 ± 2.91	[53]
**135**	Cirsilineol	206.78 ± 58.12	351.71 ± 9.72	[39]
**136**	*cis*-ε-viniferin	212.91 ± 5.87	86.65 ± 2.23	[53]
**137**	Coumestrol	415.52 ± 18.81	495.03 ± 2.99	[49]
**138**	Daidzein	145.15 ± 2.36	8.58 ± 0.94	[49]
**139**	Dieckol	1.18 ± 0.02	1.61 ± 0.08	[57]
**140**	Dioxinodehydroeckol	29.97 ± 4.52	34.60 ± 1.95	[57]
**141**	Eckol	2.64 ± 0.04	22.78 ± 2.15	[57]
**142**	Eriodictyol	64 ± 0.2	10.4 ± 0.3	[60]
**143**	*Epi*-calyxin B	27.3 ± 0.8	16.6 ± 1.7	[54]
**144**	*Epi*-calyxin H	89.2 ± 12.4	6.0 ± 0.2	[54]
**145**	Isorhamnetin	43.88 ± 2.26	141.17 ± 12.3	[39]
**146**	Katsumadainol A_1_	22.0 ± 4.9	9.4 ± 0.3	[54]
**147**	Katsumadainol A_11_	90.9 ± 14.3	3.1 ± 0.2	[54]
**148**	Katsumadainol A_12_	47.7 ± 0.4	5.8 ± 0.0	[54]
**149**	Katsumadainol A_13_	74.2 ± 4.1	6.2 ± 0.2	[54]
**150**	Katsumadainol A_14_	55.0 ± 9.3	11.2 ± 1.6	[54]
**151**	Katsumadainol A_15_	106.3 ± 1.9	11.5 ± 0.6	[54]
**152**	Katsumadainol A_16_	119.8 ± 1.5	7.6 ± 0.0	[54]
**153**	Katsumadainol A_2_	62.9 ± 2.9	7.2 ± 0.2	[54]
**154**	Katsumadainol A_3_	38.4 ± 0.8	22.0 ± 1.6	[54]
**155**	Katsumadainol A_5_	72.3 ± 13.1	17.3 ± 3.2	[54]
**156**	Katsumadainol A_6_	94.0 ± 15.1	20.6 ± 4.1	[54]
**157**	Katsumadainol A_7_	71.5 ± 9.3	23.8 ± 5.4	[54]
**158**	Luteolin	136.3	94.6	[58]
**159**	Mongolicin C	14.65 ± 0.06	4.79 ± 0.23	[56]
**160**	Moracin I	21.3 ± 1.5	49.5 ± 2.8	[55]
**161**	Moracin M	65.60 ± 3.15	16.54 ± 1.00	[56]
**162**	Moracin O	66.47 ± 1.75	45.17 ± 2.15	[56]
**163**	Moracin P	56.31 ± 1.09	55.04 ± 2.05	[56]
**164**	Moracin S	49.24 ± 1.60	13.48 ± 1.11	[56]
**165**	Morin	33 ± 2.4	4.48 ± 0.04	[61]
**166**	Morusalbin A	6.07 ± 0.12	4.53 ± 0.30	[56]
**167**	Morusalbin B	6.77 ± 0.81	5.07 ± 0.34	[56]
**168**	Morusalbin C	9.67 ± 0.08	5.40 ± 0.45	[56]
**169**	Morusalbin D	1.90 ± 0.12	3.55 ± 0.03	[56]
**170**	Mulberrofuran G	0.57 ± 0.04	1.67 ± 0.02	[59]
**171**	Mulberrofuran H	78.96 ± 3.62	12.51 ± 0.57	[56]
**172**	Mulberrofuran K	8.49 ± 0.27	5.91 ± 0.80	[56]
**173**	Mulberro-furan L	32.89 ± 1.38	10.76 ± 0.55	[56]
**174**	Paeonilactiflorol	27.23 ± 2.25	13.57 ± 1.56	[53]
**175**	Quercetin	20.35 ± 7.56	58.93 ± 6.69	[39]
**176**	*trans*-gnetin H	27.81 ± 2.74	14.39 ± 8.08	[53]
**175**	Tsaokoflavanol K	80.4 ± 3.2	5.2 ± 1.0	[62]
**178**	Tsaokoflavanol L	73.0 ± 0.5	37.5 ± 4.3	[62]
**179**	Tsaokoflavanol M	101.0 ± 4.3	58.0 ± 0.3	[62]
**180**	Tsaokoflavanol O	134.7 ± 21.3	108.5 ± 9.2	[62]
**181**	Tsaokoflavanol P	359.4 ± 14.9	15.1 ± 2.4	[62]
**182**	Tsaokoflavanol A	238.2 ± 1.0	9.0 ± 0.6	[62]
**183**	Tsaokoflavanol B	114.9 ± 0.1	7.7 ± 0.8	[62]
**184**	Tsaokoflavanol C	155.9 ± 1.8	35.8 ± 9.3	[62]
**185**	Tsaokoflavanol E	356.1 ± 0.1	20.0 ± 0.6	[62]
**186**	Tsaokoflavanol F	56.4 ± 5.0	5.6 ± 2.1	[62]
**187**	Tsaokoflavanol H	151.7 ± 0.1	13.3 ± 6.4	[62]
**188**	Tsaokoflavanol J	75.1 ± 2.4	26.8 ± 0.1	[62]
**189**	Yunanensin A	5.26 ± 0.21	3.64 ± 0.35	[56]
**190**	Ugonin J	0.6 ± 0.2	3.9 ± 0.2	[60]
**191**	2-(3,4-dihydroxyphenyl)-6-((2,2-dimethyl-6-methylenecyclohexyl)methyl)-5,7-dihydroxychroman-4-one	2.1 ± 0.5	7.4 ± 0.3	[60]
**192**	Ugonin S	7.3 ± 0.4	19.2 ± 0.5	[60]
**193**	Ugonin L	4.4 ± 0.2	19.3 ± 0.6	[60]
**194**	Ugonin M	2.7 ± 0.3	10.4 ± 0.3	[60]
**195**	Ugonin U	3.1 ± 0.7	32.9 ± 0.7	[60]

**Table 9 molecules-26-04818-t009:** Prenylated phenolic compounds.

Compound		IC_50_ (µM)		Reference
		PTP1B	α-Glucosidase	
**196**	(*E*)-4-isopentenyl-3,5,2″,4″-tetrahydroxystilbene	9.72 ± 0.83	11.65 ± 0.37	[56]
**197**	3′-*O*-Methyl-5′-*O*-methyldiplacone	3.8 ± 0.3	78.9 ± 2.1	[63]
**198**	3′-*O*-methyldiplacol	4.9 ± 0.5	17.8 ± 1.1	[63]
**199**	3′-*O*-methyldiplacone	3.9 ± 0.3	18.4 ± 0.9	[63]
**200**	4′-*O*-methyldiplacol	8.2 ± 0.6	25.8 ± 1.2	[63]
**201**	4′-*O*-methyldiplacone	7.8 ± 0.6	19.6 ± 1.1	[63]
**202**	5′-Geranyl-5,7,2″,4″-tetrahydroxy-flavone	15.53 ± 0.60	8.70 ± 0.81	[56]
**203**	6,8-Diprenylgenistein	2.3 ± 0.1	16.3 ±1.8	[64]
**204**	6-geranyl-3,3′,5,5′,7-pentahydroxy-4′-methoxyflavane	6.6 ± 0.5	2.2 ± 0.2	[63]
**205**	6-geranyl-3′,5,5′,7-tetrahydroxy-4′-methoxyflavanone	5.9 ± 0.4	6.5 ± 0.5	[63]
**206**	Albanin A	79.42 ± 1.98	18.31 ± 0.46	[56]
**207**	Glycyuralin H	5.9 ± 0.8	20.1 ± 1.6	[64]
**208**	Kuwanon C	31.24 ± 2.78	12.04 ± 0.53	[56]
**209**	Kuwanon G	2.26 ± 0.03	2.35 ± 0.03	[59]
**210**	Macrourin G	2.52 ± 0.03	3.61 ± 0.24	[56]
**211**	Mimulone	1.9 ± 0.1	30.7 ± 1.5	[63]
**212**	Morachalcone A	31.61 ± 0.31	11.85 ± 0.30	[56]
**213**	2′-*O*-demethylbidwillol B	9.2 ± 1.5	8.6 ± 0.1	[64]
**214**	Glyurallin A	8.3 ± 1.5	0.3 ± 0.0	[64]

**Table 10 molecules-26-04818-t010:** Caffeoylquinic Acid.

Compound		IC_50_ (µM)		Reference
		PTP1B	α-Glucosidase	
**215**	1,5-Dicaffeoylquinic acid	16.05 ± 1.45	146.06 ± 0.07	[39]
**216**	3,4-Dicaffeoylquinic acid	2.60 ± 0.24	128.07 ± 1.67	[39]
**217**	3,5-Dicaffeoylquinic acid	2.02 ± 0.46	217.40 ± 5.45	[39]
**218**	4,5-Dicaffeoylquinic acid	3.21 ± 0.23	229.94 ± 1.32	[39]
**219**	Methyl-3,5-di-*O*-caffeoylquinic acid	2.99 ± 0.42	86.95 ± 4.10	[39]

**Table 11 molecules-26-04818-t011:** Alkaloids.

Compound		IC_50_ (µM)		Reference
		PTP1B	α-Glucosidase	
**220**	3-Formyl-1-hydroxycarbazole	4.36 ± 0.09	87.36 ± 0.69	[66]
**221**	Clauraila B	23.89 ± 0.22	76.64 ± 0.69	[66]
**222**	Clausenaline E	32.86 ± 0.17	20.97 ± 0.63	[66]
**223**	Clausenaline F	28.42 ± 0.26	168.75 ± 1.87	[66]
**224**	Clausenanisine A	0.58 ± 0.05	3.28 ± 0.16	[66]
**225**	Clausenanisine B	0.87 ± 0.06	8.27 ± 0.42	[66]
**226**	Clausenanisine C	28.79 ± 0.16	9.38 ± 0.23	[66]
**227**	Clausenanisine D	38.48 ± 0.32	12.37 ± 0.62	[66]
**228**	Clausenanisine E	27.96 ± 0.15	96.17 ± 1.28	[66]
**229**	Clausenanisine F	2.47 ± 0.09	176.32 ± 1.09	[66]
**230**	Clausine I	3.96 ± 0.07	16.78 ± 0.45	[66]
**231**	Clausine Z	5.39 ± 0.12	17.28 ± 0.36	[66]
**232**	Clausines B	31.45 ± 0.32	16.83 ± 0.78	[66]
**233**	Clauszoline M	26.37 ± 0.16	143.57 ± 1.02	[66]
**234**	Clauszoline N	24.43 ± 0.25	15.79 ± 0.57	[66]
**235**	Dihydromupamine	15.26 ± 0.26	62.89 ± 0.82	[66]
**236**	Euchrestifoline	1.28 ± 0.07	11.96 ± 0.29	[66]
**237**	Kurryame	27.93 ± 0.28	192.23 ± 0.78	[66]

**Table 12 molecules-26-04818-t012:** Others.

Compound		IC_50_ (µM)		Reference
		PTP1B	α-Glucosidase	
**238**	Capillin	31.90 ± 0.15	332.96 ± 1.44	[67]
**239**	Capillinol	159.99 ± 0.39	464.53 ± 2.69	[67]
**240**	Dioxinodehydroeckol	29. 97 ± 4.52	34.60 ± 1.95	[57]
**241**	Eckol	2.64 ± 0.04	22.78 ± 2.15	[57]
**242**	Phlorofucofuroeckol-A	0.56 ± 0.10	1.37 ± 0.05	[57]
**243**	Phloroglucinol	55.48 ± 1.85	141.18 ± 13.1	[57]
**244**	*p*-propoxybenzoic acid	14.8 ± 0.9	10.5 ± 0.8	[38]
**245**	Sargachromenol	11.80 ± 3.35	42.41 ± 3.09	[68]
**246**	Sargaquinoic acid	14.15 ± 0.02	96.17 ± 3.48	[68]
**247**	Suffruticosol A	86.26 ± 8.47	15.57 ± 2.14	[53]
**248**	Suffruticosol B	53.93 ± 2.88	10.82 ± 2.96	[53]
**249**	Suffruticosol C	131.24 ± 3.93	45.10 ± 4.81	[53]
**250**	Tuberosin	183.95 ± 2.27	28.07 ± 3.10	[49]
**251**	Tyramine	188.06 ± 3.21	273.23 ± 5.65	[43]

**Table 13 molecules-26-04818-t013:** Distribution frequency of IC_50_ values for PTP1B and α-glucosidase.

IC_50_ (µM)	*n*° of Molecules *	
	PTP1B	α-Glucosidase
0.1–10	84 (34.2%)	65 (26.5%)
10–20	31 (12.6%)	51 (20.8%)
20–30	29 (11.8%)	21 (8.6%)
30–40	20 (8.1%)	14 (5.7%)
40–50	6 (2.4%)	14 (5.7%)
50–60	13 (5.3%)	11 (4.6%)
60–650	63 (25.6%)	69 (28.1%)

* Total number of analyzed molecules = 250.

**Table 14 molecules-26-04818-t014:** Mechanism of action of dual PTP1B/α-glucosidase inhibitors.

Compound	Inhibition Type	
	PTP1B	α-Glucosidase
Alaternin	Competitive	Mixed-type inhibition
Albanol B	Mixed-type inhibition	Mixed-type inhibition
Albasin B	Non competitive	Competitive
Dieckol	Non competitive	Competitive
Murasalbin D	Non competitive	Competitive
7-Phloroeckol	Non competitive	Non competitive
Macrourin G	Non competitive	Competitive
6-geranyl-3,3′,5,5′,7-pentahydroxy-4′-methoxyflavane	Mixed-type inhibition	Non competitive
Phlorofucofuroeckol-A	Non competitive	Non competitive
Ugonin J	Competitive	Non competitive
Ugonin M	Mixed-type inhibition	Mixed-type inhibition

**Table 15 molecules-26-04818-t015:** Calculated IC_50_ ratio.

Number of Compounds	* α-Glucosidase IC_50_/PTP1B IC_50_
110 (44.7%)	1
35 (14.2%)	2
20 (8.0%)	3
19 (7.6%)	4
13 (5.2%)	5
50 (20.3%)	>5

* Compounds were pooled based on the value of the calculated IC_50_ ratio. The most represented group includes compounds showing a similar IC_50_ values for both targets. For this reason, these showed a ratio value comprised between 0.5 and 1. The other groups include compounds showing a lower affinity for α-glucosidase in respect to PTP1B.

**Table 16 molecules-26-04818-t016:** Physicochemical properties of the sixty most potent MDLs obtained from PUBCHEM database.

Parameters	Average Values of the Top Sixty Compounds	Lipinski’s Criteria
miLogP ^1^	4.9 ± 2	0–5
hydrogen bond acceptors ^2^	6.4 ± 4	<10
hydrogen bond donors ^3^	3.8 ± 2	<5
TPSA ^4^	112 ± 59	<140 Å
rotatable bonds ^5^	5.2 ± 4	<10–20
Molecular weight	432 ± 127 Da	<500 Da

^1^ Octanol/Water Partition Coefficient (partition coefficients are useful in estimating the distribution of drugs within the cells: hydrophobic drugs, showing high octanol–water partition coefficients, are mainly distributed to lipid bilayers of cells, whereas hydrophilic compounds possess a low octanol–water partition coefficients and accumulate in cytoplasm of the cells). ^2^ Number of hydrogen bond acceptors (the number of hydrogen bond acceptors in the structure). ^3^ Number of hydrogen bond donors in the structure (the number of hydrogen bond donors in the structure.). ^4^ TPSA represents the surface sum over all polar atoms in a molecule. ^5^ Any single-order, non-ring bond where atoms on either side of the bond are in turn bound to nonterminal heavy (i.e., non-hydrogen) atoms.

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
