# Peer review of "Natural α-Glucosidase and Protein Tyrosine Phosphatase 1B Inhibitors: A Source of Scaffold Molecules for Synthesis of New Multitarget Antidiabetic Drugs"

_molecules, 2021, doi:10.3390/molecules26164818_

Round 1

Reviewer 1 Report

The following points need to be addressed (please refer to the attached examined copy)

  • SAR is required
  • There is no discussion for any of the tables from 1-12, which are very suitable for SAR, which also not mentioned, the same tables missing the natural sources
  • The same tables (1-12, all of them) containing repetition and incorrect nomenclatures
  • Abbreviation section is required
  • The ligands of all figures (check 1-4) need careful revision, the info posted there are inappropriate
  • Some of the titles need careful revision (e.g.1.5, 3).
  • Other language was spotted in the review, careful revision required
  • References need to be corrected according to the Journal style

Author Response

Question: SAR is required There is no discussion for any of the tables from 1 12,
which are very suitable for SAR, which also not mentioned, the same tables missing the natural sources
Response: as suggested by referee, the natural sources of each molecule were cited at the beginning of each table. Moreover, according to referre’s suggestion for each compound category a structure activity relationship (SAR) was carried out. Each analysis was inserted below t he corresponding t ables (the text was highlighted in red).

Question : The same tables (1-12, all of them) containing repetition and incorrect
nomenclatures
Response: we apologize for the repetitions and errors present in the tables. To
overcome this problem, we have numbered each compound and ordered the names of the molecules in alphabetical order. Duplicates have been eliminated from the Tables 1-12

Questions: The legends of all figures (check 1 4) need careful revision, the info
posted there are inappropriate
Response: according to referee’s suggestion, we revised legends of figures

Question: Some of the titles need careful revision (e.g.1.5, 3).
Response: in accordance with the referee's suggestions, we have corrected the
chapter titles

Question: Other language Other language was spotted in the review, careful revision required
Response: following the referee's suggestion, we double--checked all the text  and corrected the errors 

Question: References need to be corrected according to the Journal style 
Response:  all references have been checked and formatted following the Journal's indications

Reviewer 2 Report

The article is interesting and contains very important information, nevertheless the writing and information of the tables is not suitable.

Alphabetical order (or by bibliographical reference) of the contained information is suggested in the tables.

There are tables with repeated information  

The numerical description of the authors suggests different place from work

Author Response

Question: The article is interesting and contains very important information, nevertheless the writing and information of the tables is not suitable.
Response: we apologize for the repetitions and errors present in the tables. We checked and rationalized the information reported in the Tables

Question: Alphabetical order (or by bibliographical reference) of the contained information is suggested in the tables.
Response: as suggested by referee, we have numbered and organized the compounds in alphabetical order

Question: There are tables with repeated information
Response: we have checked all compounds reported in the Table to remove duplicates

Question: The numerical description of the authors suggests different place from work
Response: the numerical description of the authors has been corrected. All authors show the same affiliation

Reviewer 3 Report

Reviewer’s comments and suggestions

The current study was discussing natural compounds known as α-Glucosidase and Protein Tyrosine Phosphatase 1B (PTP1B) dual-inhibitors as starting scaffolds for new multitarget antidiabetic drugs for T2D treatment. The approach of writing the review was poor and it would be limited for the researcher to completely understand the manuscript. The review paper consisted of data’s which were not mentioned that how the author can get it in the paper. Papers look missing information such as how they collected the data and which software they used (mainly material and method section) although other parts are okay and understandable. That’s why I am providing a chance to the author to update their manuscript and resubmit it.

Below are the comments needed to be incorporated into the manuscript.

  1. Line 20, “the most diffused” Not appropriate
  2. Figure needs to be improved, think about it
  3. Line 68-69 Please explore the sentence
  4. Line 137-138 what does it mean
  5. Please provide a reference for line 172-174
  6. Different kinds of PPAR dual-target drugs have been 184 produced in the last years [14] “ delete the line”
  7. Line 216 “absorption” not absorbtion
  8. Line 227-228 needed a reference
  9. Line 238-239 I am not convincing the sentence
  10. The line is this correct “compound among them showed IC50 values for α-glucosidase and PTP1B of 10, 96 and 268 13.45 μM, respectively and a good selectivity for PTP1B”
  11. How it is possible without stimulation? “these compounds show a good insulin-mimetic activity, enhancing phosphorylation 274 levels of Akt in the absence of insulin stimulation”
  12. All Table titles needed to be explained and How the author selected it
  13. Line 339 “emerges from the data reported in this review is that the natural world can 339
  14. provide a large number of” How the author collected the data, there was no information as such and one more thing I am curious about the data collection in the paper (table 14)
  15. The reader should understand your data completely if they know how you have explored the result (table 15) 
  16. Better to explain step by step Properties of sixty most potent MDLs obtained from PUBCHEM database
  17. For figure 6 How the author did the experiment, which tool they have used
  18. What does the author mean here “modulating the increase of postprandial blood glucose levels? This finding indicated that Phlorofucofuroeckol A can be used to reduce postprandial hyperglycemia in diabetic patients “
  19. Typo error line 556 and 602
  20. Line 696 I did not find in the paper
  21. Line 703 How is it possible if the author did not discuss important things so that other researcher can follow the paper

Author Response

1) Question : Papers look missing information such as how they collected the data and which software they used
Response: as suggested by referee, information about the method used to
collect data were reported in the chapter 2 (pag 10).
2) Question : Line 20, “the most diffused” Not appropriate
Response: the statement has been corrected
3) Question : Figure needs to be improved, think about it
Response: As suggested, the text of Figures 1-3 has been changed
4) Question : Line 68-69 Please explore the sentence
Response: as suggested, the sentence has been corrected
5) Question : Line 137 138 what does it mean
Response: this phrase had been changes
6) Question: Please provide a reference for line 172-174
Response: as suggested, a reference has been added
7) Question : Different kinds of PPAR dual target drugs have been 184 produced
in the last years [14] “ delete the line”
Response: as suggested, the line 184 has been deleted
8) Question : Line 216 “absorption” not absorbtion
Response: the mistake has been corrected
9) Question : Line 227-228 needed a reference
Response: as suggested, the reference ha s been added
10) Question : Line 238-239 I am not convincing the sentence
Response: as suggested, the sentence has been modified
11) Question: The line is this correct “compound among them showed IC50 values for α glucosidase and PTP1B of 10, 96 and 268 13.45 μM , respectively and a good selectivity for PTP1B”
Response: as suggested, we corrected this sentence
12) Question: How it is possible without stimulation? “these compounds show a
good insulin mimetic activity, enhancing phosphorylation 274 levels of Akt in
the absence of insulin stimulation”
Response:  one hypothesis about molecular mechanisms promoting the
incerase of Akt phosphorylation levels has been added.
13) Question: All Table titles needed to be explained and How the author selected it
Response: compounds were categorized basing on their structural properties.
By using this principle, all compounds have been divided in twelve different
classes.
14) Question: Line 339 “emerges from the data reported in this review is that the natural world can 339 provide a large number of” How the author collected the data, there was no information as such and one more thing I am curious about the data collection in the paper (table 14)
Response: Information reported in the table 1 12 were collected analysing
literature data. T he table did not show chemical structure of each molecule
because this would require a lot of space. In the same way, we analyse all paper
collected looking for information about the inhibition mechanism of selected
compounds. Unfortunately, only few paper contained detailed kinetic analyses.
All data collected were reported in the table 14
15) Question: The reader should understand your data completely if they know
how you have explored the result (table 15)
Response: as suggested by referee, we added so me sentences below Table 15 to made more simple the meaning of data showed
16) Question : Better to explain step by step Properties of sixty most potent MDLs obtained from PUBCHEM database
Response: as suggested, we added an explanation of the meaning of each
parameter
17) Question:  For figure 6 How the author did the experiment, which tool they have used 
Response: as suggested we explained which tool was used for this experiment
18) Question: What does the author mean here “modulating the increase of
postprandial blood glucose levels? This finding indicated that
Phlorofucofuroeckol A can be used to reduce postprandial hyperglycemia in
diabetic patients “
Response:  we changes this sentence to made more clear the in vivo effects of
compound.
19) Question: Typo error line 556 and 602e 556 and 602
Response: the typo errors have been corrected
20) Question:  Line 696 I did not find in the paper
Response: this line is now visible
21) Question: Line 703 How is it possible if the author did not discuss important
things so that other researcher can follow the paper
Response: taking into account all corrections made, we completely revised the “Conclusion” section

Round 2

Reviewer 1 Report

Dear Authors

Thanks for the comprehensive revision made, much appreciated. few small errors found need to be addressed such as:

1- compound 8 table 1, please use the common name given in the article.

2- all names containing O, R, S,  (such as compds 17, 18, 19,   ...etc) should be italics, in case of D, this need to be capital but small font (say 10)

3- the plant names should fellow the standard way of writing, i.e italic (please correct the name in line 450) and many other places all over the MS

4- table 4, please check compounds 65,..etc for a, b (which should be alpha and beta), epi, compound 73 (italic), other R and S also italics. make sure the names in the text also are correct.

5- compound 86 and 90 same corrections (E also italics, a, b are alpha, beta, p, is italic), make sure from the text as well.

6-table 6, all Z should be italics (and the text)

7- compound 118, 119, 142 also the same

8-compound 190 correct as follow: -methylenecyclohexyl)methyl)-5,7-dihydroxychroman-4-one

9- table 9 and 10 also, all O's, should be italics.

10- the new numbers under the section "Mechanism of action and in vivo activity of some     " are very confusing, please give the compounds the same numbers mentioned in the tables, and if not mentioned before continue the number where you ended, i.e 251, 252,...etc.

Author Response

We thanks referees 1 for his suggestions.
The new manuscript contain all corrections suggested by referee. Below, you can find the point by point response to referee's comments.

Reviewer 3 Report

No more comments. 

Author Response

We thanks referees 3 for his suggestions.